# Generation of inner ear hair cells by direct lineage conversion of primary somatic cells

Louise Menendez[1,2,3], Talon Trecek[1,2], Suhasni Gopalakrishnan[1,2,3], Litao Tao[1,2], Alexander L Markowitz[3,4], Haoze V Yu[1,2], Xizi Wang[1,2], Juan Llamas[1,2], Chichou Huang[5], James Lee[5], Radha Kalluri[3,4], Justin Ichida[1,2,3]*, Neil Segil[1,2,4]*

[1]Department of Stem Cell and Regenerative Medicine, University of Southern California, Los Angeles, United States; [2]Eli and Edythe Broad Center, University of Southern California, Los Angeles, United States; [3]Zilkha Neurogenetic Institute, University of Southern California, Los Angeles, United States; [4]USC Caruso Department of Otolaryngology – Head and Neck Surgery, University of Southern California, Los Angeles, United States; [5]DRVision Technologies, Bellevue, United States

**Abstract** The mechanoreceptive sensory hair cells in the inner ear are selectively vulnerable to numerous genetic and environmental insults. In mammals, hair cells lack regenerative capacity, and their death leads to permanent hearing loss and vestibular dysfunction. Their paucity and inaccessibility has limited the search for otoprotective and regenerative strategies. Growing hair cells in vitro would provide a route to overcome this experimental bottleneck. We report a combination of four transcription factors (*Six1, Atoh1, Pou4f3,* and *Gfi1*) that can convert mouse embryonic fibroblasts, adult tail-tip fibroblasts and postnatal supporting cells into induced hair cell-like cells (iHCs). iHCs exhibit hair cell-like morphology, transcriptomic and epigenetic profiles, electrophysiological properties, mechanosensory channel expression, and vulnerability to ototoxin in a high-content phenotypic screening system. Thus, direct reprogramming provides a platform to identify causes and treatments for hair cell loss, and may help identify future gene therapy approaches for restoring hearing.

*For correspondence:
ichida@usc.edu (JI);
nsegil@med.usc.edu (NS)

## Introduction

Hearing loss is the most common sensory deficit with estimates of around 466 million people affected worldwide (*WHO, 2019*). Loss of sensory hair cells of the inner ear is the primary cause of sensorineural hearing loss (*Bohne and Harding, 2000*; *Hinojosa et al., 2001*; *Géléoc and Holt, 2014*; *Wong and Ryan, 2015*). The highly structured sensory epithelium of the inner ear, known as the organ of Corti, develops from a post-mitotic, pro-sensory domain established in the developing cochlear duct between embryonic days E12.5 and E14.5 in mice (*Ruben and Sidman, 1967*; *Lowenheim et al., 1999*; *Chen and Segil, 1999*; *Matei et al., 2005*; *Lee et al., 2006*). These post-mitotic cells are the progenitors for sensory hair cells and their adjacent supporting cells (*Fekete et al., 1998*; *Kelley, 2006*; *Driver et al., 2013*). Sensory hair cells function as the essential mechanoreceptors that convert sound vibrations into electrical signals, which are then transmitted to the brain via the spiral ganglion neurons that innervate the hair cells (*Géléoc and Holt, 2003*).

Sensory hair cells are located in both the auditory and vestibular portions of the inner ear (*Figure 1A*). The hair cells within the organ of Corti are precisely arranged into one row of inner hair cells and three rows of outer hair cells, interdigitating with a variety of supporting cells; inner border, inner phalangeal, pillar cells, Deiters' cells and Hensen's cells (*Figure 1A*). Hair cells are susceptible

**eLife digest** Worldwide, hearing loss is the most common loss of sensation. Most cases of hearing loss are due to the death of specialized hair cells found deep inside the ear. These hair cells convert sounds into nerve impulses which can be understood by the brain. Hair cells naturally degrade as part of aging and can be damaged by other factors including loud noises, and otherwise therapeutic drugs, such as those used in chemotherapy for cancer. In humans and other mammals, once hair cells are lost they cannot be replaced.

Hair cells have often been studied using mice, but the small number of hair cells in their ears, and their location deep inside the skull, makes it particularly difficult to study them in this way. Scientists are seeking ways to grow hair cells in the laboratory to make it easier to understand how they work and the factors that contribute to their damage and loss. Different cell types in the body are formed in response to specific combinations of biological signals. Currently, scientists do not have an efficient way to grow hair cells in the laboratory, because the correct signals needed to create them are not known.

Menendez et al. have now identified four proteins which, when activated, convert fibroblasts, a common type of cell, into hair cells similar to those in the ear. These proteins are called Six1, Atoh1, Pou4f3 and Gfi1. Menendez et al. termed the resulting cells induced hair cells, or iHCs for short, and analyzed these cells to identify those characteristics that are similar to normal hair cells, as well as their differences. Importantly, the iHCs were found to be damaged by the same chemicals that specifically harm normal hair cells, suggesting they are useful test subjects.

The ability to create hair cells in the laboratory using more easily available cells has many uses. These cells can help to understand the normal function of hair cells and how they become damaged. They can also be used to test new drugs to assess their success in preventing or reversing hearing loss. These findings may also lead to genetic solutions to curing hearing loss.

to degeneration by a variety of genetic mutations and environmental stressors, such as exposure to loud noise, ototoxic drugs including cancer chemotherapy and aminoglycoside antibiotics, aging and over 200 known syndromic and non-syndromic genetic loci conferring predispositions to hearing loss (*Matsui et al., 2004*; *Cheng et al., 2005*; *Bodmer, 2008*; *Langer et al., 2013*; *Atkinson et al., 2015*; *Wong and Ryan, 2015*; *Vaisbuch and Santa Maria, 2018*). In mammals hearing and balance are dependent on the maintenance of hair cells present at birth (*Groves, 2010*; *Géléoc and Holt, 2014*), since hair cells do not spontaneously regenerate (*Roberson and Rubel, 1994*; *Chardin and Romand, 1995*; *Forge et al., 1998*), and so their death leads to lifelong hearing loss and balance disorders. In contrast, non-mammalian species, such as birds and reptiles, are able to spontaneously regenerate lost hair cells from existing supporting cells, leading to full functional recovery (*Corwin and Cotanche, 1988*; *Ryals and Rubel, 1988*; *Stone and Cotanche, 2007*; *Brignull et al., 2009*).

Transcription factors regulate the temporal and spatial patterns of gene expression within the cells of complex tissues, establishing cell fate, and ultimately determining their morphological and functional properties (*Lemon and Tjian, 2000*; *Levine and Tjian, 2003*; *Zhang et al., 2004*). Within the inner ear, expression of *Atoh1*, a bHLH class transcription factor (*Lo et al., 1991*; *Ross et al., 2003*) is both necessary and sufficient for the induction of sensory hair cells in the embryonic and neonatal cochlea, and ultimately plays an integral role in initiating the hair cell gene expression program (*Bermingham et al., 1999*; *Zheng and Gao, 2000*; *Woods et al., 2004*; *Kelly et al., 2012*; *Chonko et al., 2013*; *Cai et al., 2013*; *Ryan et al., 2015*; *Scheffer et al., 2015*; *Stojanova et al., 2016*; *Costa et al., 2017*). However, previous studies have shown that *Atoh1* expression alone is not sufficient to induce hair cell differentiation in somatic cells (*Izumikawa et al., 2008*; *Costa et al., 2015*; *Abdolazimi et al., 2016*), or mature supporting cells of the organ of Corti (*Kelly et al., 2012*; *Liu et al., 2012b*).

The paucity and inaccessibility of primary inner ear hair cells have limited the identification of effective otoprotective and regenerative strategies. Recent studies have demonstrated the in vitro formation of hair cells from murine pluripotent stem cells and human embryonic stem cells by directed differentiation (*Oshima et al., 2010*; *Koehler et al., 2013*; *Li et al., 2003*; *Ronaghi et al.,*

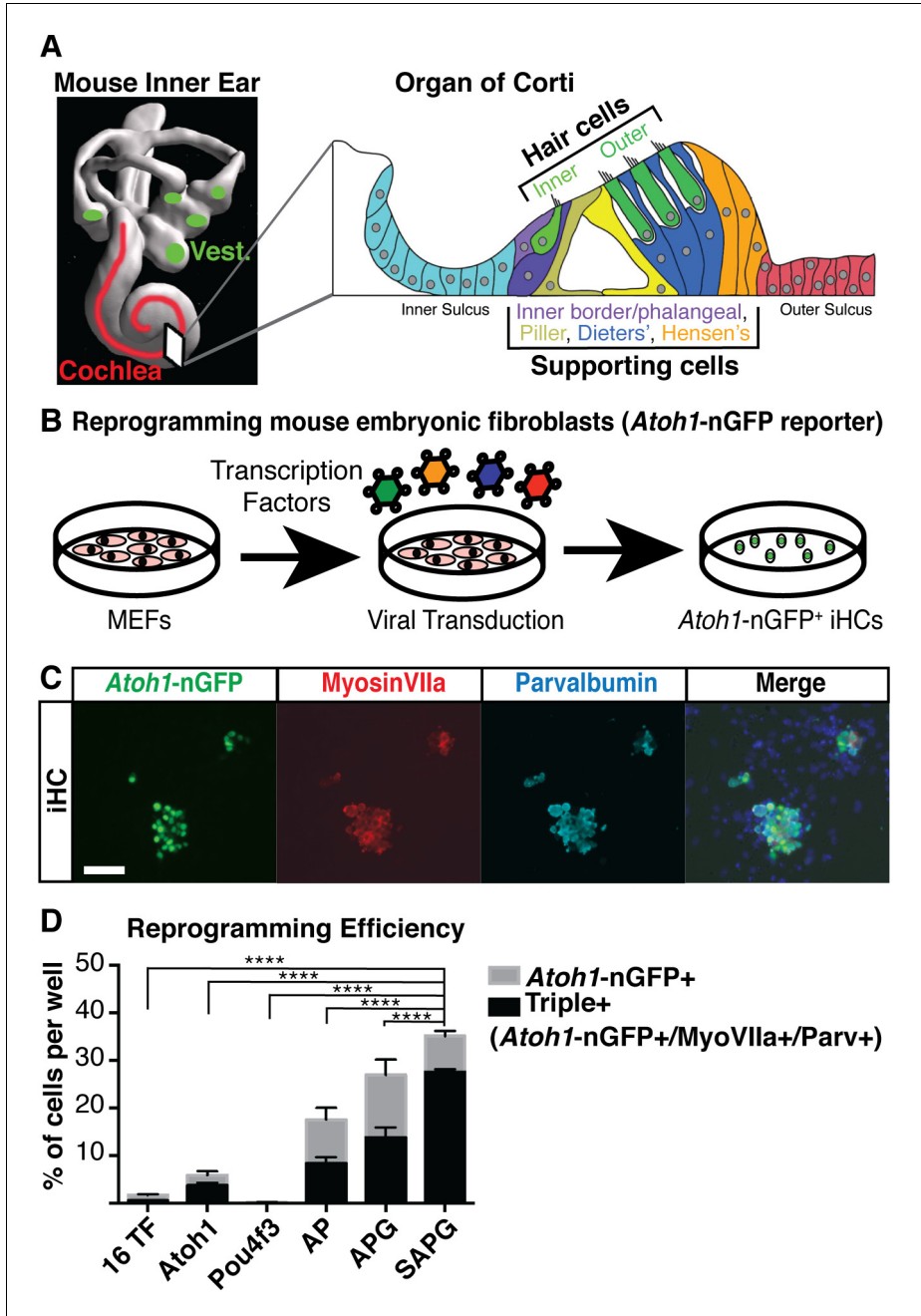

**Figure 1.** Overexpression of Six1, Atoh1, Pou4f3 and Gfi1 is capable of activating hair cell markers in mouse embryonic fibroblasts. (**A**) Diagram of the mouse inner ear shows the vestibular system (green) and the cochlea of the auditory system (red). Cross section through one turn of the cochlea shows organization in the organ of Corti as a mosaic of sensory hair cells (one row of inner hair cells and three rows of outer hair cells) interdigitated by various supporting cell populations labeled from left to right (Inner border/phalangeal, Pillar, Deiters' and Hensen's). Space filling model care of Steven Raft. (**B**) Schematic of experimental design for transcription factor mediated reprogramming. Mouse embryonic fibroblasts (MEFs) were isolated from *Atoh1*-nGFP transgenic reporter mice. MEFs were plated at a density of 5000 cells per well of a 96 well plate, infected with retroviral transcription factors and allowed to reprogram for 14 days prior to analysis. (**C**) Images of MEFs reprogrammed with *Six1, Atoh1, Pou4f3*, and *Gfi1* (SAPG) fixed at 14 days post infection (dpi). *Atoh1*-nGFP reporter activation (green) and immunostaining for anti-MyosinVIIa (red) and anti-Parvalbumin (cyan). Scale bar represents 50 um in length. (**D**) All quantification was performed at 14 dpi. Reprogramming efficiency was calculated as the number of *Atoh1*-nGFP positive cells divided by the 5000 MEFs plated per well. Reporter activation and immunostaining for anti-MyosinVIIa and anti-Parvalbumin was used to quantify triple positive cells (*Atoh1*-nGFP+/MyosinVIIa+/

*Figure 1 continued on next page*

*Figure 1 continued*

Parvalbumin+). A = *Atoh1*, P=*Pou4* f3, G = *Gfi1*, S = *Six1*. The combination SAPG gave 35% (± 1.8) reprogramming efficiency and 78% (± 1.9) of *Atoh1*-nGFP+ cells were triple positive. Statistics shown are for the comparison of triple positive cells in each condition. (N = 3 independent experiments per condition, n = 3 replicates per condition per experiment; mean ± SEM; one-way ANOVA *p<0.05, **p<0.01, ***p<0.001, ****p<0.0001).

The online version of this article includes the following figure supplement(s) for figure 1:

**Figure supplement 1 .** Overexpression of Six1, Atoh1, Pou4f3 and Gfi1 is capable of activating hair cell markers in mouse embryonic fibroblasts.

2014), or in a combination of directed differentiation to an ectodermal, non-neural, placodal cell type, followed by transcription factor induction to a hair cell-like state (*Costa et al., 2015*). However, these elegant approaches require three-dimensional culture conditions that complicate high-throughput studies, for instance screening for otoprotectants. In contrast to morphogen-based directed differentiation of pluripotent stem cells, transcription factor (TF) -mediated lineage conversion of somatic cells enables the rapid production of neurons and other cell types in microtiter plates with ≥96 wells, allowing the reproducibility and homogeneity required for high-throughput phenotypic screening (*Xu et al., 2015*; *Babos et al., 2019*). Thus, the identification of a transcription factor cocktail that can convert somatic cells into sensory hair cells could enable screening for new otoprotective targets. Moreover, delivery of such a cocktail in vivo would enable regenerative medicine strategies for hair cell replacement in situ, which have thus far been ineffective (*Izumikawa et al., 2005*; *Richardson and Atkinson, 2015*; *Roccio et al., 2015*).

To this end, we have identified a cocktail of four transcription factors, *Six1*, *Atoh1*, *Pou4f3*, and *Gfi1* (SAPG), capable of converting mouse embryonic fibroblasts, adult tail tip fibroblasts, and postnatal mouse supporting cells into induced hair cells (iHCs). iHCs are highly similar to primary hair cells in terms of global gene expression and chromatin accessibility profiles, morphological features, and electrophysiological properties. In addition, we established a robotic imaging platform with automated analysis to track iHC survival and show that like primary hair cells, iHCs are selectively sensitive to gentamicin toxicity. These findings show that iHCs make a valuable in vitro model to study hair cell regeneration, maturation, function and susceptibility to ototoxins.

## Results

### Direct reprogramming of MEFs with *Six1*, *Atoh1*, *Pou4f3* and *Gfi1* activates key hair cell markers

To identify a group of TFs needed to convert somatic cells into induced hair cells, we analyzed the transcriptome of postnatal day 1 (P1) cochlear hair cells that had been FACS-purified from a transgenic mouse expressing GFP in nascent hair cells under the control of an *Atoh1* 3' enhancer (*Lumpkin et al., 2003*). We compared the primary P1 cochlear hair cell transcriptome to a reference transcriptome of the FACS-purified GFP-negative cells from the same organ of Corti preparations (*Figure 1—figure supplement 1A*). We identified 16 candidate TFs that were highly enriched in P1 hair cells (*Atoh1*-nGFP+), some of which are known to have essential roles in hair cell development (*Li et al., 2003*; *Wallis et al., 2003*; *Qian et al., 2006*; *Hume et al., 2007*; *Ahmed et al., 2012*; *Chonko et al., 2013*; *Liu et al., 2014a*; *Cai et al., 2015*; *Scheffer et al., 2015*).

Using retroviral delivery, we transduced the TFs into mouse embryonic fibroblasts (MEFs) from the *Atoh1*-nGFP reporter mouse (*Figure 1B*). MEFs transduced with a control virus (dsRed) did not express the *Atoh1*-nGFP transgene after 14 days (*Figure 1—figure supplement 1B*). In contrast, overexpression of all 16 TFs led to *Atoh1*-nGFP activation in 1.7% (± 0.3) of MEFs at 14 days post infection (*Figure 1—figure supplement 1C*). Reprogramming efficiency was calculated as a percent of *Atoh1*-nGFP-positive MEFs out of the starting MEF number (5000 cells per well). This result indicated that within this initial group were individual transcription factors, or combinations thereof, able to reprogram MEFs to a hair cell-like state. The low level of reprogramming efficiency is expected when large numbers of factors are infected simultaneously, since only a subset of factors is expected to infect any given cell (*Phan and Wodarz, 2015*; *Mistry et al., 2018*), and since using large

numbers of factors, and/or virus, is likely to challenge cellular transcription/translational machinery, thus further reducing efficiency (*Babos et al., 2019*).

To identify the TFs critical for the *Atoh1*-nGFP reporter activation in MEFs, we tested the efficiency of *Atoh1* and each of the other 15 TFs separately (*Figure 1—figure supplement 1C*). We observed that *Atoh1* alone activated the *Atoh1*-nGFP reporter in 5.8% (± 1.5) of starting MEFs, while *Pou4f3*-alone only did so in 0.15% (± 0.03) of the starting MEFs (*Figure 1—figure supplement 1C*). None of the other 14 factors alone activated the *Atoh1*-nGFP reporter. We then tested the reprogramming efficiency of *Atoh1* in combination with each of the other 15 TFs (*Figure 1—figure supplement 1C*). The most significant reporter activation came from a combination of *Atoh1* and *Pou4f3*, which provided 17.5% (± 4.4) reprogramming efficiency (*Figure 1—figure supplement 1D*). We then tested the addition of each remaining individual factor to the combination of *Atoh1* and *Pou4f3* (AP) (*Figure 1—figure supplement 1E*). *Gfi1* combined with AP (APG) increased the reporter activation to 26.9% (± 5.6) reprogramming efficiency (*Figure 1—figure supplement 1E*). A subsequent round of addition of individual TFs to this three-factor combination showed that the addition of *Six1* to *Atoh1*, *Pou4f3*, and *Gfi1* (SAPG) further increased the reporter activation to reach 35.2% (± 1.8) reprogramming efficiency (*Figure 1—figure supplement 1F*). Addition of the remaining individual factors to the cocktail of SAPG did not increase reprogramming efficiency (*Figure 1—figure supplement 1G*).

Since *Atoh1* is expressed in other cell types and lineages (*Klisch et al., 2011*; *Kim et al., 2014*; *Ostrowski et al., 2015*), we performed immunostaining for MyosinVIIa and Parvalbumin, two additional markers that are more specific to a hair cell fate (*Eybalin and Ripoll, 1990*; *Demêmes et al., 1993*; *Pak and Slepecky, 1995*; *Hasson et al., 1997*; *Sahly et al., 1997*; *Richardson et al., 1997*; *Richardson et al., 1999*; *Boëda, 2002*). The majority of SAPG-transduced cells that activated *Atoh1*-nGFP also expressed MyosinVIIa and Parvalbumin (78.4% ± 1.9) (*Figure 1C,D*). Overall, SAPG transduction activated *Atoh1*-nGFP with a 35% efficiency, and nearly 80% of all *Atoh1*-nGFP positive cells also expressed MyosinVIIa and Parvalbumin (*Figure 1D*). In contrast, only 50% of the *Atoh1*-nGFP+ cells generated by AP or APG expressed MyosinVIIa and Parvalbumin, indicating that most *Atoh1*-nGFP+ cells generated from these alternative cocktails were not hair cell-like (*Figure 1D*). Our results support the importance of *Six1*, *Atoh1*, *Pou4f3* and *Gfi1* in direct reprogramming of somatic cells to a hair cell-like state, with high efficiency, purity, and reproducibility.

## Induced hair cells resemble primary juvenile mouse hair cells transcriptionally

Direct lineage reprogramming relies on the forced expression of transcription factors to induce a molecular rewiring of the transcriptional programs that characterize specialized cells (*Takahashi and Yamanaka, 2006*; *Takahashi et al., 2007*). This involves both upregulation of the target cell-specific genes, in this case of primary cochlear hair cells, and downregulation of the starting cell-specific genes, in this case of MEFs. To assay the extent to which induced hair cells replicate the mouse primary cochlear hair cell gene expression program, we performed RNA-sequencing on FACS-purified *Atoh1*-nGFP+ cells generated by overexpression of *Six1*, *Atoh1*, *Pou4f3*, and *Gfi1* (SAPG) at 14 days post infection (dpi)(hereafter referred to as iHCs). We compared the gene expression of the iHCs to FACS-purified *Atoh1*-nGFP+ primary cochlear hair cells at postnatal day 1 (P1; hereafter referred to as P1 HCs), and MEFs infected with a control retrovirus expressing a fluorescent protein (dsRed; hereafter referred to as MEFs). We categorized the gene expression in iHCs as either 'successfully reprogrammed', 'not-reprogrammed' or 'inappropriately expressed' and divided the categories into those genes that are normally expressed in P1 HCs, but not in MEFs (P1 HC genes, black bar), and those that are normally expressed in MEFs, but not in primary hair cells (MEF genes, black bar) (*Figure 2A*). From this analysis we determined that the iHCs had transcriptionally activated a hair cell-like signature by successfully upregulating 64% of P1 HC genes, while simultaneously becoming distinct from the starting MEF population by successfully downregulating 69% of MEF genes (*Figure 2A*). These percentages are comparable to those achieved in the TF-induced direct conversion of MEFs into spinal motor neurons, as well as those attained in MEF-to-cardiomyocyte and hepatocyte-to-neuron direct conversion (*Ieda et al., 2010*; *Marro et al., 2011*; *Gopalakrishnan et al., 2017*; *Ichida et al., 2018*). These results suggest that iHCs largely resemble primary P1 cochlear hair cells at the transcriptional level.

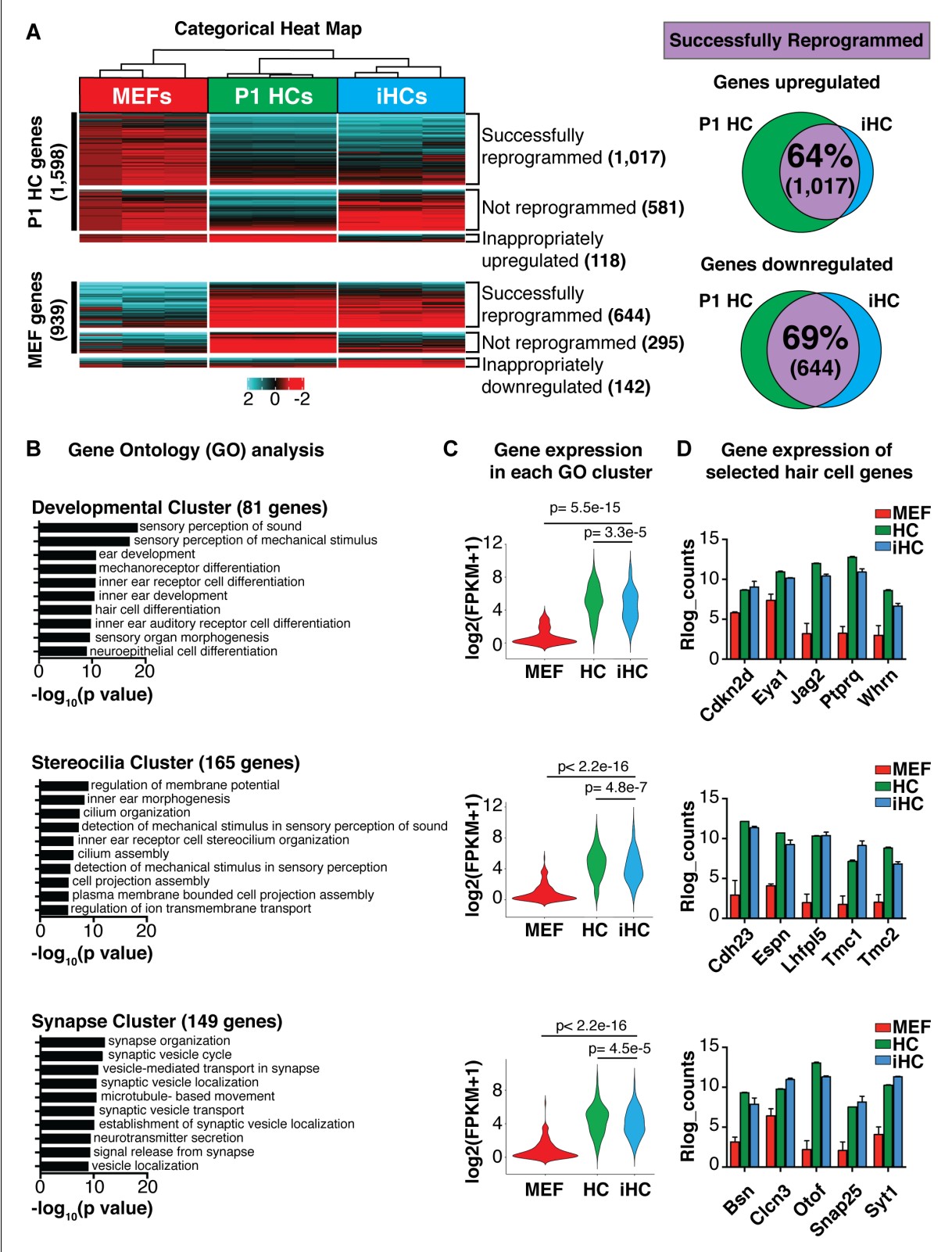

**Figure 2.** Transcriptional analysis of induced hair cells (iHCs) indicates expression profile similarity to primary cochlear hair cells. (A) Categorical heat-map comparing gene expression (RNA-seq) of mouse embryonic fibroblasts (MEFs), P1 cochlear hair cells (P1 HCs), and induced hair cells (iHCs). (n = 3 replicates for P1 HCs, n = 3 replicates for iHCs, n = 3 replicates for MEFs). Venn diagrams show percent of correctly reprogrammed genes. 64% of uniquely expressed P1 HC genes are correctly upregulated during reprogramming, and 69% of inappropriately expressed MEF genes are
*Figure 2 continued on next page*

*Figure 2 continued*

downregulated. (B) Gene Ontology analysis (GO-terms) of successfully upregulated genes in iHCs categorized into three relevant gene clusters: Development, Stereocilia and Synapse. (C) RNA expression of the gene set associated with each GO Cluster. Violin plots show relative expression of genes in each of the three clusters: Development (81 genes), Stereocilia (165 genes) and Synapse (149 genes). Gene lists in *Figure 2—figure supplement 1*. (D) Gene expression (Rlog counts) in MEFs, iHCs and P1 HCs for selected hair cell-enriched genes in each of the three clusters. All genes are significantly upregulated in iHCs compared to MEFs with p<0.05 and FDR < 0.01. (n = 3 replicates for P1 HCs, n = 3 replicates for iHCs, n = 3 replicates for MEFs; mean ± SEM).

The online version of this article includes the following source data and figure supplement(s) for figure 2:

**Source data 1.** Gene Ontology gene lists Gene Ontology (GO) analysis showed that the genes successfully upregulated during reprogramming were revealed as three clusters of GO terms: development-related GO terms, stereocilia-related GO terms, and synapse-related GO terms.

**Figure supplement 1.** Transcriptional analysis of induced hair cells (iHCs) indicates expression profile similarity to primary cochlear hair cells.

Nonetheless, a number of genes did not respond to the SAPG group of transcription factors used for reprogramming. Of the 1506 genes expressed in P1 HCs, but not MEFs, 36% were not successfully upregulated in the iHCs, and 118 genes were inappropriately upregulated. Of the 939 genes that are expressed in MEFs, but not in P1 HCs, and thus need to be downregulated during reprogramming, 31% failed to downregulate, and 142 genes were inappropriately downregulated. PCA analysis of bulk RNA-seq data from MEFs, compared to either primary P1 HCs or iHCs show the relative difference between these populations (*Figure 2—figure supplement 1B*).

Gene Ontology (GO) analysis showed that the genes successfully upregulated during reprogramming were significantly enriched for 'sensory perception of sound' and 'detection of mechanical stimulus', which revealed as three clusters of genes (*Figure 2B*). The first cluster was enriched for development-related GO terms such as 'inner ear receptor cell development', 'mechanoreceptor differentiation', and 'hair cell differentiation' (*Figure 2B*). The second cluster was enriched for stereocilia-related GO terms such as 'plasma membrane bound cell projection assembly', 'cilium organization' and 'cilium movement' (*Figure 2B*). The third cluster was enriched for synaptic signaling GO terms such as 'establishment of synaptic vesicle localization', 'synaptic vesicle cycle' and 'neurotransmitter secretion' (*Figure 2B*). These three clusters of GO terms were used to generate cluster-specific gene sets driving the GO designation (*Figure 2C*, *Figure 2—source data 1*). Expression levels of each GO cluster-specific gene set, in each cell type, were plotted to visualize the statistically significant iHC divergence from MEFs, and iHC convergence towards a P1 HC expression profile (*Figure 2C*). This analysis also revealed a significant difference in the level of gene expression (p<0.05) between iHCs and P1 HCs (*Figure 2C*). This difference may be explained by the maturity level of the iHCs as well as the presence of the residual MEF transcriptional profile. However, further investigation of the Rlog count values for several key genes in each GO cluster demonstrated that the iHCs efficiently upregulated important hair cell genes including *Whirlin* (*Whrn*) involved in cell polarity, *Cadherin23* (*Cdh23*) and *Espin* (*Espn*) important for stereocilia organization and functionality, as well as *Bassoon* (*Bsn*) and *Otoferlin* (*Otof*) required for synaptic scaffolding and synaptic vesicle signaling (*Figure 2D*). After looking at the expression of key hair cell genes, we determined that the iHCs had also activated all of the initial transcription factors included in the set of 16 candidate factors, with the exception of *Zfp503* (*Figure 2—figure supplement 1D*). Together, the RNA sequencing results indicate that the iHCs generated by *Six1, Atoh1, Pou4f3*, and *Gfi1* (SAPG) overexpression are capable of repressing most of the initial MEF gene signature, while simultaneously adopting a gene expression signature similar to primary P1 HCs.

Interestingly, the reprogramming transcription factors (SAPG), are normally expressed in both cochlear and utricular differentiating hair cells. We compared postnatal day one cochlear hair cells (P1 cHCs) and utricular hair cells (P1 uHCs) to iHCs. PCA analysis shows that iHCs have not become more similar to one of the two primary hair cell populations (*Figure 2—figure supplement 1A*). The PCA also demonstrates that at this early developmental time point the P1 cHCs and P1 uHCs are themselves immature, and have similar transcriptional profiles. Nonetheless, the expression profile of iHCs has drastically shifted away from the MEF signature and towards an immature hair cell-like signature.

## Induced hair cells are distinct from other *Atoh1* dependent lineages

Expression of *Atoh1* is necessary and sufficient for hair cell differentiation in the context of the inner ear primordium (*Bermingham et al., 1999*; *Chen et al., 2002*; *Woods et al., 2004*; *Chonko et al., 2013*; *Cai et al., 2013*), however several other lineages including cerebellar granule cell progenitors (*Klisch et al., 2011*), Merkel cells (*Ostrowski et al., 2015*), and the secretory cell lineage of the gut (*Kim et al., 2014*) rely on *Atoh1* expression for differentiation. To characterize the specificity of our reprogramming to the hair cell-like state, we analyzed RNA sequencing data from FACS-purified iHCs relative to other *Atoh1*-dependent lineages including cerebellar granule precursors (CGP), secretory cells of the gut (GUT), and Merkel cells (MC). Principle component analysis (PCA) indicated that iHCs were more similar to primary P1 cochlear hair cells (P1 HC) than to either cerebellar granule cell precursors (CGP) (*Figure 3A*), secretory cells of the gut (GUT) (*Figure 3B*), or Merkel cells (MC) (*Figure 3C*), showing that they established a hair cell-specific transcriptional program and have not adopted the transcriptional profile of other *Atoh1*-dependent lineages.

As an additional test, we compared iHCs to the other *Atoh1*-dependent lineages using Gene Set Enrichment Analysis (GSEA) (*Subramanian et al., 2005*). By comparing the transcriptomes in MEFs, P1 hair cells (HC), P1 Cerebellar granule precursors (CGP), adult gut secretory cells (GUT), and P1 Merkel cells (MC), we defined groups of genes as part of a specific signature for each cell type (*Figure 3—source data 1*). The GSEA program identified gene-lists exclusive to each cell type; these gene-lists included only genes which were not expressed in any two cell types. We calculated Normalized Enrichment Scores (NES) (*Subramanian et al., 2005*) for each cell type in comparison to iHCs. The largest NES was for the comparison of iHCs to P1 hair cells (HC), indicating an enrichment for the HC gene signature, while showing lower enrichment scores, and even negative enrichment scores, for the other cell type comparisons (*Figure 3D*). Thus, reprogramming with SAPG establishes a hair cell-like transcriptional program without adopting the transcriptional profiles of other *Atoh1*-dependent lineages.

## Chromatin accessibility profile of induced hair cells resembles that of primary cochlear hair cells

Chromatin structure controls the accessibility of genes for either activation or repression in response to developmental and environmental signaling (*Volpi et al., 2000*; *Kozubek et al., 2002*; *Goetze et al., 2007*; *Buenrostro et al., 2015b*; *Chen et al., 2016*; *Sijacic et al., 2018*), and as such, is an important regulator of cell type-specific gene expression. We used an Assay of Transposase Accessible Chromatin (ATAC) sequencing (*Buenrostro et al., 2015a*; *Chen et al., 2016*) to analyze the regions of open/accessible chromatin in MEFs (MEF peaks), primary P1 hair cells (P1 HC peaks), and iHCs (*Figure 4A*). As in our analysis of gene expression (*Figure 2*), we characterized the open chromatin regions into those that are present in primary P1 HCs, but not in MEFs (P1 HC peaks, black bar), and those that are open in MEFs, but not primary HCs (MEF peaks, black bar), as analyzed by ATAC-seq accessibility (*Figure 4A*). We defined these groups, as in *Figure 2*, as either 'successfully reprogrammed', 'not reprogrammed', and 'inappropriately opened/closed' (i.e. not matching either P1 HC peaks or MEF peaks).

The iHCs show robust opening of de novo distal element regions of the chromatin that are open in P1 HCs, as well as large-scale chromatin closing in regions of the genome that were accessible in the starting MEF population. The hair cell-appropriate changes in chromatin accessibility of iHCs are also accompanied by a proportion of inappropriate opening or closing of chromatin regions. Of the 13,390 peaks present uniquely in P1 HCs, 73% were successfully opened during reprogramming, while 27% were not opened during reprogramming, and an additional 18,084 peaks opened inappropriately in iHCs (*Figure 4A*). Conversely, of the 26,847 peaks unique to MEFs, 84% of peaks were successfully closed during reprogramming, 16% were not closed during reprogramming, and an additional 833 peaks were inappropriately closed during reprogramming (*Figure 4A*).

Since most distal accessible elements are not active enhancers in a given cell type (*Heintzman et al., 2007*), we analyzed the H3K27ac-state of the distal elements present in P1 HCs, a marker of active enhancers (*Creyghton et al., 2010*; *Figure 4B*). These results show that most of the enhancers identified in P1 HCs are opened in iHCs. Global enhancer targets have not been analyzed in these cell types due to small numbers, so we arbitrarily assigned putative gene targets to each P1 HC enhancer by identifying the closest transcriptional start site. This is expected to identify

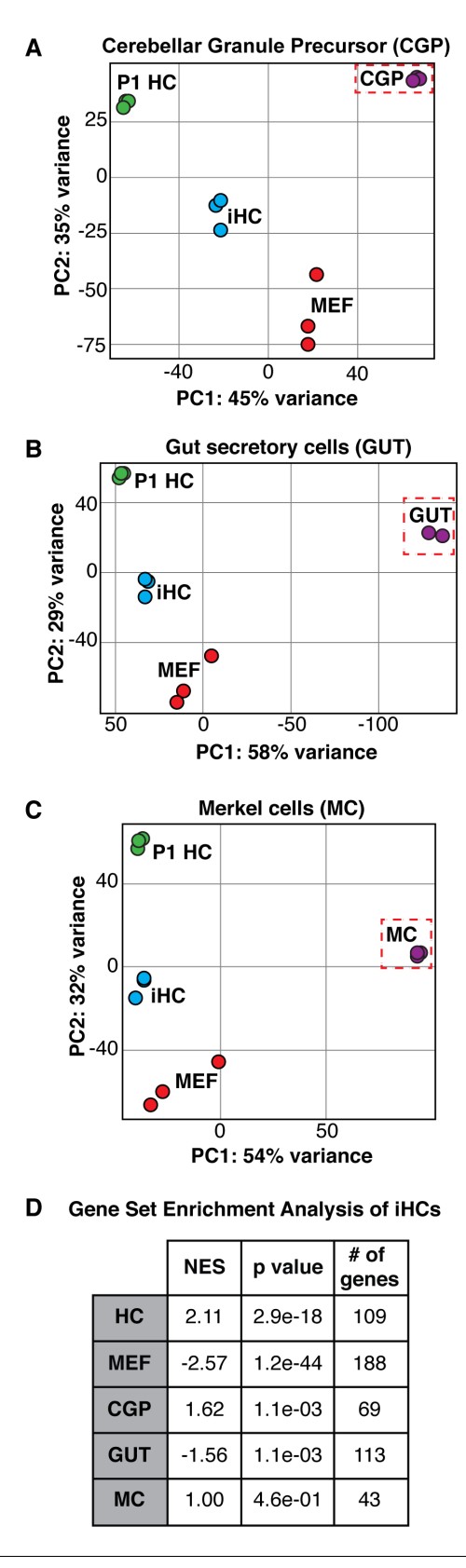

**Figure 3.** The expression profile of induced hair cells resembles primary hair cells and is distinct from other

27–47% of genuine targets, based on chromosome conformation capture experiments performed in other cell types (*Sanyal et al., 2012*). Based on our RNA-seq data, we found that the genes defined as putative targets of P1 HC-specific enhancers had significantly higher expression in both P1 HCs and iHCs compared to MEFs (*Figure 4C*).

To visualize the ATAC-seq and ChIP-seq data at specific loci we used the Integrative Genomics Viewer (IGV) (*Robinson et al., 2011*). We chose four known hair cell loci, *Pou4f3*, *Mreg*, *Rasd2*, and *Barhl1*, that exemplify the changes in chromatin structure at H3K27ac-defined enhancers between P1 HCs, iHCs and MEFs (*Figure 4D*). These results indicate that robust and hair cell-appropriate global changes in chromatin accessibility accompany the large shift in the transcriptional profile of iHCs.

## *Six1*, *Atoh1*, *Pou4f3*, and *Gfi1* are capable of reprogramming postnatal and adult somatic cells

We have used mouse embryonic fibroblasts (MEFs) as a starting cell type for our reprogramming efforts in the experiments described thus far. However, MEFs are an embryonic and relatively heterogeneous cell population (*Singhal et al., 2016*). To test the ability of the SAPG transcription factors to reprogram a mature somatic cell from a heterologous lineage, we chose as proof-of-principle, to reprogram *Atoh1*-nGFP transgenic adult tail tip fibroblasts (TTFs). We virally transduced TTFs with *Six1*, *Atoh1*, *Pou4f3*, and *Gfi1* (*Figure 5A*). While TTFs are known to be infected by retrovirus at a lower efficiency than MEFs (*Liu et al., 2011*; *Lalit et al., 2016*), SAPG activated the *Atoh1*-nGFP reporter in adult tail tip fibroblasts (*Figure 5A*). In addition, SAPG catalyzed the expression of Myosin-VIIa and Parvalbumin, indicating that as in MEFs, these transcription factors can convert adult tail tip fibroblasts into *Atoh1*-nGFP+/MyosinVIIa+/Parvalbumin+ iHCs (*Figure 5A,B*).

Supporting cells are an attractive target for gene therapy approaches to hair cell regeneration, due to their known role in regeneration in non-mammalian vertebrates (*Corwin and Cotanche, 1988*; *Ryals and Rubel, 1988*; *Stone and Cotanche, 2007*; *Brignull et al., 2009*), their having a common progenitor with hair cells (*Fekete et al., 1998*; *Kelley, 2006*; *Driver et al., 2013*), and their survival in long-deafened mice (*Oesterle and Campbell, 2009*). Although hair cells do not regenerate in the mature mammalian cochlea, perinatal supporting

*Figure 3 continued*

*Atoh1*-dependent lineages. (**A–C**) Principle component analysis showing the differences in transcriptional profiles between P1 cochlear hair cells (P1 HCs), induced hair cells (iHCs), control mouse embryonic fibroblasts (MEFs), and cerebellar granule precursor cells (CGP), secretory cells of the gut (GUT), and Merkel cells (MC), respectively. (**D**) Gene Set Enrichment Analysis (GSEA) comparing iHC expression profile to the unique gene sets of MEFs, HCs, CGPs, GUT cells, and MCs. Table reports normalized enrichment score (NES), p values, and number of genes in each gene set. Normalized Enrichment Score (NES) shows the strongest correlation between gene sets was between iHCs and the primary hair cell (HC) gene set. Gene signature lists in *Figure 3—source data 1*.

The online version of this article includes the following source data for figure 3:

**Source data 1.** Gene Set Enrichment Analysis gene lists Gene Set Enrichment Analysis (GSEA) (*Subramanian et al., 2005*) was used to compare the transcriptomes in MEFs, P1 hair cells (HC), P1 Cerebellar granule precursors (CGP), adult Gut secretory cells (GUT), and P1 Merkel cells (MC).

cells have been shown to have a transient ability to directly transdifferentiate into hair cells in response to Atoh1 (*Kelly et al., 2012*; *Liu et al., 2012b*), or loss of Notch-mediated lateral inhibition (*Mizutari et al., 2013*; *Maass et al., 2015*), however this potential is lost at very early postnatal stages (*White et al., 2006*; *Takebayashi et al., 2007*; *Doetzlhofer et al., 2009*; *Liu et al., 2012a*; *Cox et al., 2014*; *Bramhall et al., 2014*).

One plausible route to the in vivo regeneration of hair cells in the organ of Corti would be the conversion of mature supporting cells into hair cells in long deafened individuals. *Atoh1* alone can convert perinatal supporting cells into hair cells (*Kelly et al., 2012*; *Liu et al., 2012b*; *Yang et al., 2013*), but transdifferentiation potential decreases rapidly thereafter, such that by two weeks of age, neither *Atoh1* expression, nor induction of transdifferentiation by Notch-inhibition, can induce the transdifferentiation of supporting cells to a hair cell fate (*Maass et al., 2015*; *Jiang et al., 2016*). To determine if *Six1*, *Atoh1*, *Pou4f3*, and *Gfi1* (SAPG) are together able to convert supporting cells into hair cells from organs that had passed this transdifferentiation-permissive stage, we labeled P1 supporting cells using a transgenic cross, Lfng-CreERt2::Rosa26$^{tdTomato}$, which allows for permanent labeling of supporting cells with a tdTomato fluorescent marker (*Figure 5—figure supplement 1B*). We dissociated organs of Corti from *Lfng*-CreERt2::Rosa26$^{tdTomato}$ mice at P8, a time when induced-transdifferentiation is no longer possible, and transduced them with virus encoding *Atoh1* alone, or the combination of four factors, SAPG. Cells were infected and allowed to reprogram for two weeks before immunostaining for MyosinVIIa and Parvalbumin (*Figure 5C*). The lineage traced cells, from now on referred to as *Lfng*-tdTomato-positive P8 supporting cells (P8 SC) transduced with the SAPG produced significantly more cells that activated MyosinVIIa and Parvalbumin than *Atoh1*-transduced supporting cells (*Figure 5D*). Since the *Lfng*-tdTomato reporter is independent of the viral SAPG, the percent of triple positive (*Lfng*-tdTomato+/MyosinVIIa+/Parvalbumin+) iHCs was calculated from the total number of *Lfng*-tdTomato-positive cells per well. These results indicate that the combination of *Six1*, *Atoh1*, *Pou4f3*, and *Gfi1* can convert adult tail tip fibroblasts and P8 supporting cells into induced hair cells at a significantly greater rate than *Atoh1* alone.

## Morphological characterization of induced hair cells

Sensory hair cells have a very distinct morphology. As their name suggests, these specialized cells possess hair-like actin-based apical membrane protrusions, called stereocilia, that contain at their tips the mechanically gated ion channels required for mechanotransduction (*Kawashima et al., 2011*; *Pan et al., 2013*; *Holt et al., 2014*). Development of stereocilia involves the elaboration of a single primary, tubulin-based, cilium, known as the kinocilium, centered on a cuticular plate of F-actin filaments from which the stereocilia arise as elongated bundles of microvilli (*Cotanche and Corwin, 1991*; *Troutt et al., 1994*; *Leibovici et al., 2005*; *Wang et al., 2005*; *Tarchini et al., 2016*; *McGrath et al., 2017*).

To assess the morphological properties of iHCs we performed immunostaining at 14 days post-infection following SAPG reprogramming, which is approximately 10 days after initial *Atoh1*-nGFP detection. At this time, iHCs exhibited highly polarized F-actin staining as observed by well-defined labeling of Phalloidin-Rhodamine near the apical surface and a primary cilium that labels with antibody to acetylated tubulin, and is centered on the nascent cuticular plate (*Figure 6A*). This highly polarized pattern is reminiscent of hair cells in both the developing cochlear and vestibular systems

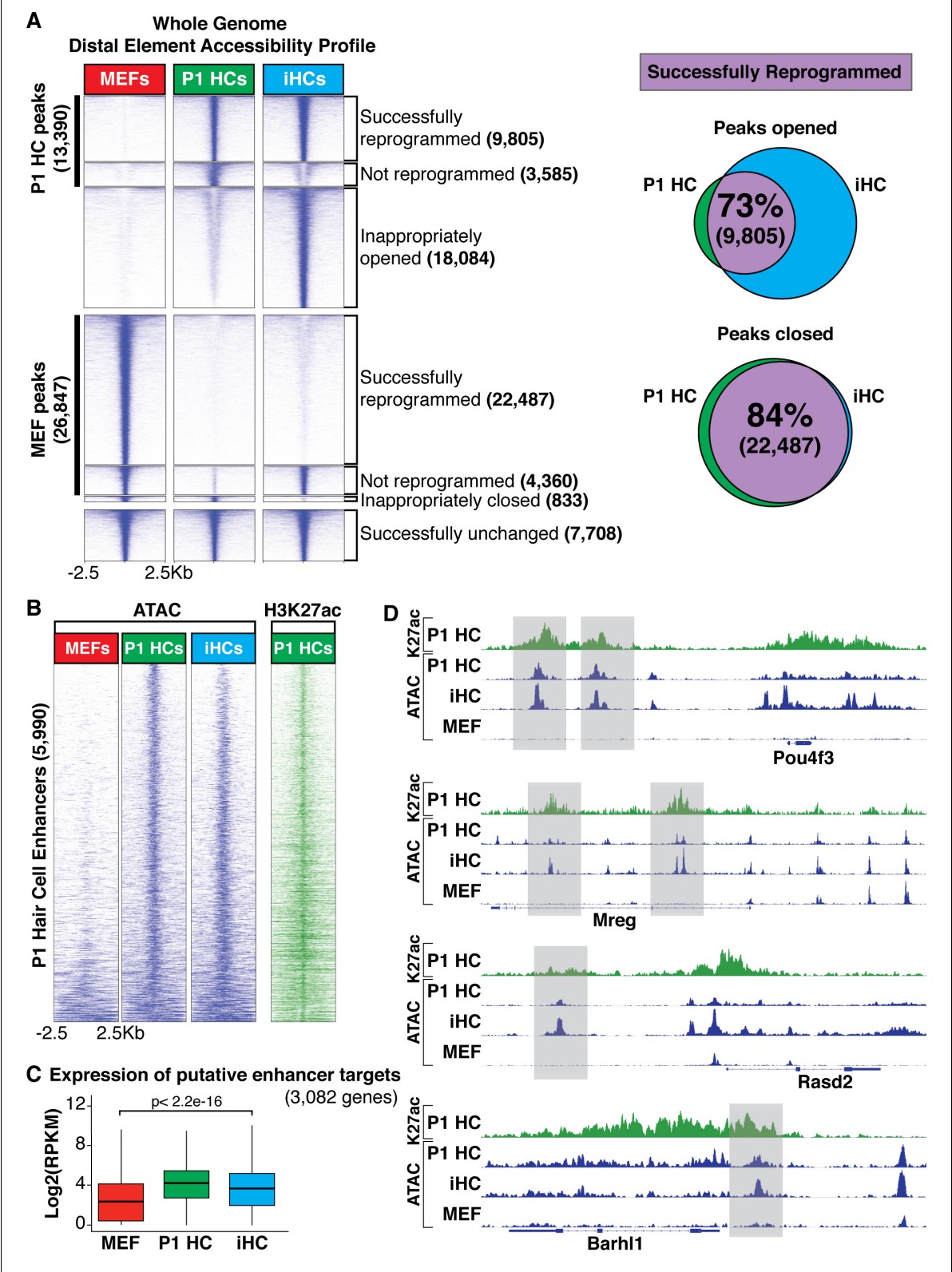

**Figure 4.** Chromatin accessibility of induced hair cells (iHCs) indicates profile similarity to primary cochlear hair cells. (**A**) Heat map comparing genome wide chromatin accessibility profiles of MEFs, P1 HCs, and iHCs. Accessibility is divided into seven clusters: 1) successfully reprogrammed HC peaks, 2) not reprogrammed HC peaks, 3) inappropriately opened peaks, 4) successfully closed MEF peaks, 5) not reprogrammed MEF peaks, 6) inappropriately closed peaks, and 7) successfully unchanged peaks. Scale of each sample column is ± 2.5 Kb from ATAC peak. Venn diagrams show percent of correctly

*Figure 4 continued on next page*

Figure 4 continued

reprogrammed chromatin regions. 73% of unique P1 HC chromatin regions are successfully opened during reprogramming, and 84% of MEF chromatin regions are successfully closed in reprogramming. (B) Heat map comparing chromatin accessibility at primary cochlear hair cell enhancers. Enhancers were identified as regions with open chromatin ATAC peaks and H3K27ac in P1 HCs. ChIP data for H3K27ac shown as green heat map and ATAC chromatin accessibility data of the respective regions shown as blue heat maps. Heat maps ordered from low to high information content. iHCs successfully open P1 HC enhancer regions that were closed in the starting MEF population. Scale of each sample column is ± 2.5 Kb from ATAC peak. (C) Expression levels (Log2(RPKM)) of 3082 putative primary hair cell enhancer targets. Enhancer targets were identified by mapping to the nearest transcription start site for each enhancer and the expression of each putative target was acquired from the RNA-seq results. iHCs significantly upregulate the expression of the putative primary hair cell enhancer targets. (D) Integrative Genomics Viewer (*Robinson et al., 2011*) tracks show primary hair cell H3K27ac profile alongside chromatin accessibility profiles of P1 HCs, iHCs and MEFs. Chromatin accessibility changes at specific hair cell enhancers for *Pou4f3, Mreg, Rasd2* and *Barhl1* are highlighted in grey boxes.

(*Cotanche and Corwin, 1991*; *Troutt et al., 1994*; *Leibovici et al., 2005*; *Wang et al., 2005*; *Tarchini et al., 2016*; *McGrath et al., 2017*).

Previous work has demonstrated that mixing dissociated cells from embryonic or perinatal organ of Corti with periotic mesenchyme, allows them to rapidly self-organize and differentiate in vitro into

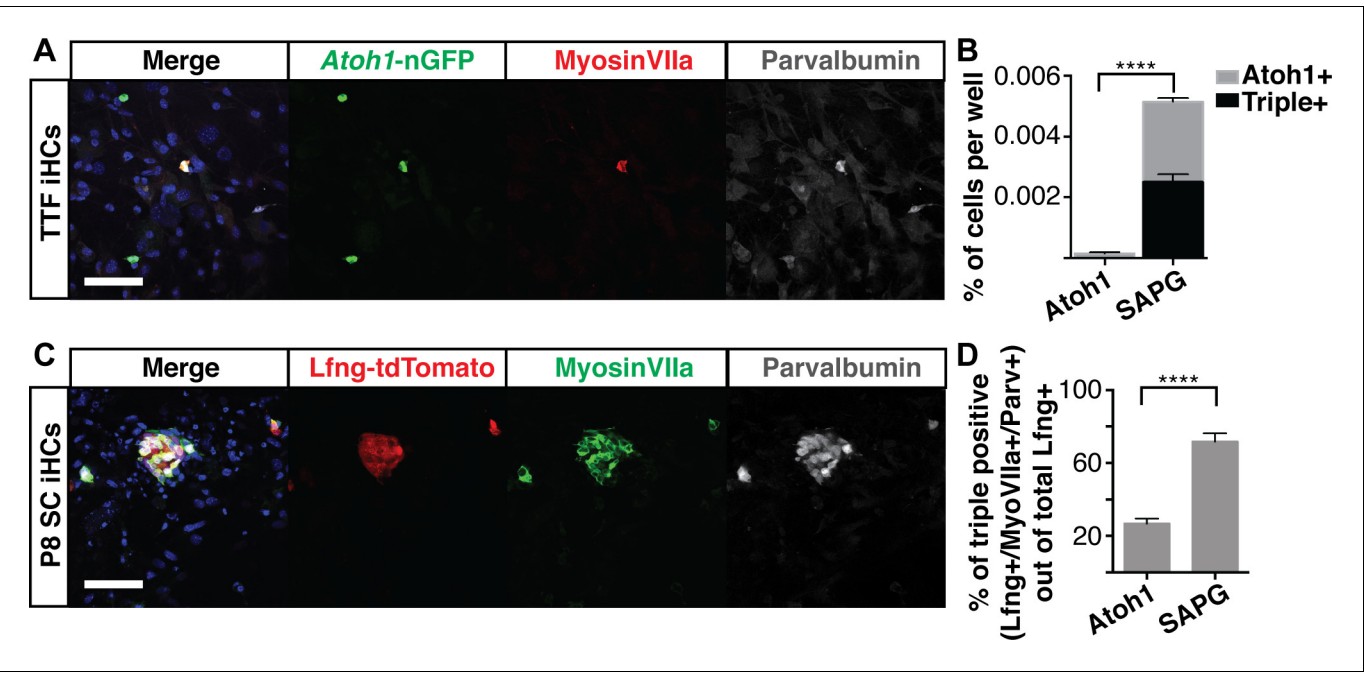

**Figure 5.** Six1, Atoh1, Pou4f3 and Gfi1 are capable of reprogramming adult cells. (A) Adult mouse tail tip fibroblasts (TTFs) were infected with SAPG and allowed to reprogram for 14 days before fixation and immunostaining. Reprogrammed TTF are able to activate the *Atoh1*-nGFP reporter and stain for anti-MyosinVIIa (red) and anti-Parvalbumin (grey). Merged image includes Hoechst nuclear stain (blue). TTFs were reprogrammed in HCM supplemented with 10% FBS and RepSox. Scale bar represents 50 um. (B) Quantification of *Atoh1*-nGFP+ cells and triple positive cells (*Atoh1*-nGFP+/ MyosinVIIa+/Parvalbumin+) in TTFs infected with *Atoh1* alone or SAPG. TTFs infected with SAPG generate significantly more *Atoh1*-nGFP+ cells and 48.6% (± 12) of the *Atoh1*-nGFP+ cells are triple positive. TTFs were reprogrammed in HCM supplemented with 10% FBS and RepSox. (N = 3 experiments, n = 3 replicates per experiment, mean ± SEM; Statistics shown for the comparison of triple positive cells in each condition; Student's t-test *p<0.05, **p<0.01, ***p<0.001, ****p<0.0001). (C) Dissociated organs of Corti from P8 transgenic mice with lineage traced supporting cells (*Lfng*-tdTomato+ SC) were infected with SAPG. Cells were reprogrammed for 14 days prior to fixation and immunostaining for anti-MyosinVIIa (green) and anti-Parvalbumin (grey). P8 *Lfng*-tdTomato+ SCs infected with SAPG are able to activate primary hair cell markers MyosinVIIa and Parvalbumin. Scale bar represents 50 um. (D) Quantification of the percent of triple positive cells (*Lfng*-tdTomato+/MyosinVIIa+/Parvalbumin+) out of the total number of *Lfng*-tdTomato+ supporting cells per well in cultures infected with *Atoh1* alone or SAPG. Presence of the *Lfng*-tdTomato reporter is independent of the viral infection. With *Atoh1* alone 26.8% (± 7) of the total *Lfng*-tdTomato+ supporting cells are able to activate the primary hair cell markers MyosinVIIa and Parvalbumin, while with SAPG 71.8% (± 12) of the total *Lfng*-tdTomato+ supporting cells are able to activate primary hair cell markers. (n = 7 replicates, mean ± SEM; Student's t-test *p<0.05, **p<0.01, ***p<0.001, ****p<0.0001).

The online version of this article includes the following figure supplement(s) for figure 5:

**Figure supplement 1.** Six1, Atoh1, Pou4f3 and Gfi1 are capable of reprogramming adult cells.

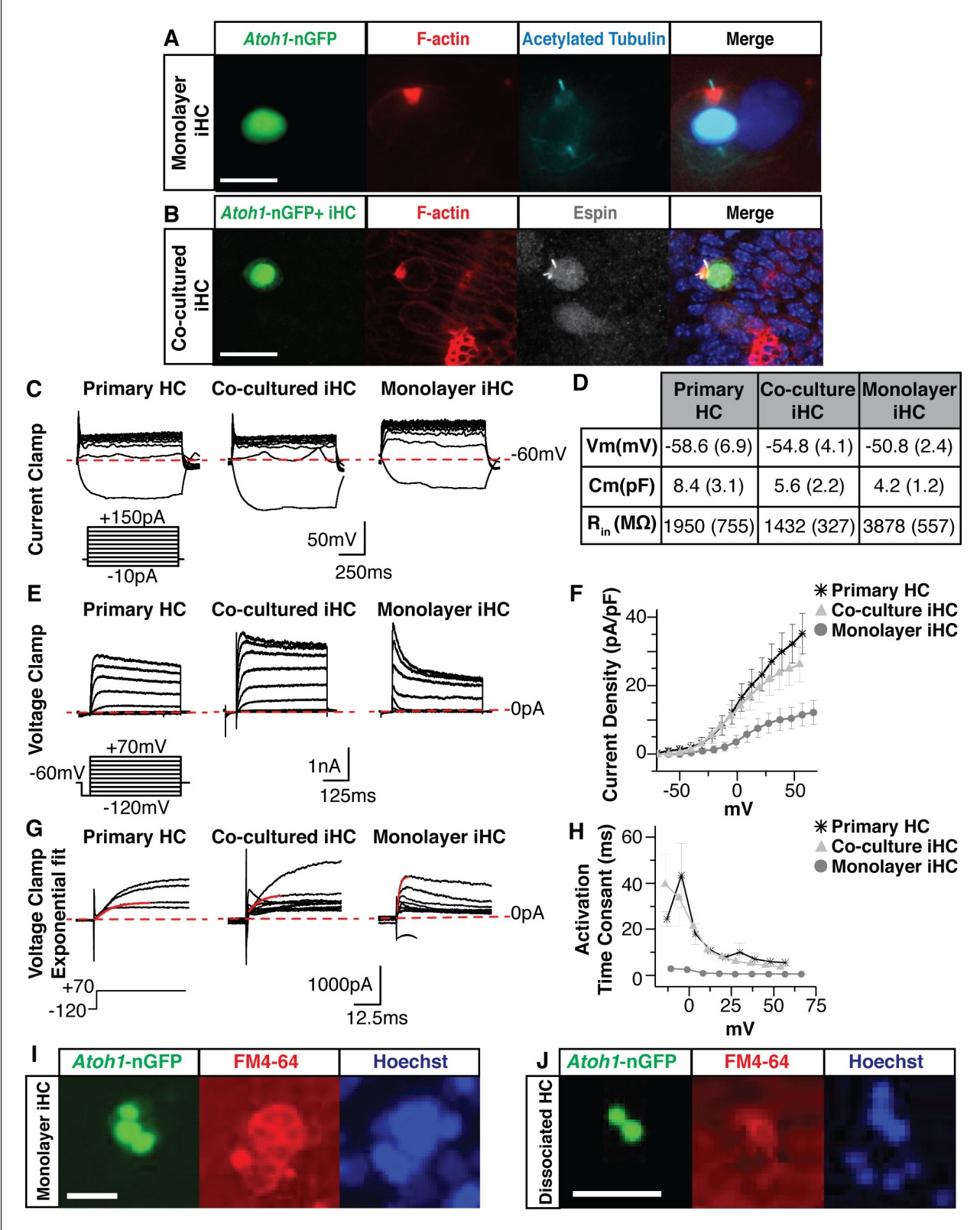

**Figure 6.** Induced hair cells demonstrate functional properties reminiscent of primary hair cells. (A) Images of monolayer-cultured iHCs show polarized F-actin by Phalloidin labeling (red) and a kinocilium by anti-acetylated Tubulin labeling (cyan). Merged image includes Hoechst nuclear stain. Scale bar represents 10 um. (B) iHCs co-cultured with dissociated primary E13.5 organs of Corti show an F-actin rich cuticular plate by Phalloidin labeling (red) and stereocilia by anti-Espin labeling (grey). Merged image includes Hoechst nuclear stain. Scale bar represents 20 um. (C) Whole cell patch clamping

*Figure 6 continued on next page*

*Figure 6 continued*

was performed on P1 HCs from a dissociated organ of Corti, co-cultured iHCs and monolayer-cultured iHCs. Results from current clamp show the change in cell voltage as a response to an applied current. Dashed red line represents −60 mV. Current clamp protocol shows steps from −10 to +150 pA in 20 pA increments. Scale bars represent 50 mV on X-axis and 250 ms on Y-axis. (D) Basic membrane properties were calculated from the current clamp data to report resting membrane potential (Vm), membrane capacitance (Cm) and input resistance (Rin). Table reports mean (SEM) for each value. (E) Results from voltage clamp shows the current output of the cell as a response to applied voltage for primary HCs, co-cultured iHCs and monolayer-cultured iHCs. Dashed red line represents 0 pA. Voltage clamp protocol shows steps from −120 to +70 mV in 10 mV increments. Scale bars represent 1 nA on X-axis and 125 ms on Y-axis. (F) IV curve plotting current density (normalized for cell size) as a function of applied voltage for primary HCs, co-cultured iHCs and monolayer-cultured iHCs. Co-cultured iHCs show similar current output to P1 primary hair cells. (G) Exponential fits to the voltage clamp traces were used to calculate the current activation time constants for primary HCs, co-cultured iHCs and monolayer-cultured iHCs. Dashed red line represents 0 pA. Solid red line shows exponential fit to outward currents when clamped from −120 mV to +70 mV. Scale bars represent 1000 pA on X-axis and 12.5 ms of Y-axis. (H) Current activation time constants reported for P1 HCs, cocultured iHCs and monolayer-cultured iHCs. Co-cultured iHCs show similar current activation kinetics to P1 HCs. (I) iHCs expressing the *Atoh1*-nGFP reporter accumulate the styryl dye FM4-64. Image taken after 30 s of incubation with FM4-64. Nuclei labeled in blue using NucBlue live dye. Scale bar represents 25 um. (J) Dissociated primary hair cells expressing the *Atoh1*-nGFP reporter accumulate the styryl dye FM4-64. Image taken after 30 s of incubation with FM4-64. Nuclei labeled in blue using NucBlue live dye. Scale bar represents 25 um.

The online version of this article includes the following figure supplement(s) for figure 6:

**Figure supplement 1.** Induced hair cells demonstrate functional properties reminiscent of primary hair cells.

epithelial island-like structures (*Doetzlhofer et al., 2004*; *White et al., 2006*). To determine if iHCs are capable of integrating appropriately into these sensory epithelial-like structures which contain both primary hair cells and supporting cells, we FACS-purified *Atoh1*-nGFP+ iHCs and mixed them with dissociated primary embryonic (E13.5) sensory epithelium, containing primary hair cells, primary supporting cells, and a portion of the surrounding periotic mesenchyme (*Figure 6—figure supplement 1A-C*). After two weeks of co-culture, iHCs contained polarized cuticular plates (F-actin) and stereocilia (espin-positive), and were found incorporated into the epithelial islands containing native hair cells and supporting cells (*Figure 6B*). Greater than 80% of iHCs that engrafted in epithelial islands exhibited highly polarized F-actin staining (data not shown). Thus, iHCs morphologically resemble primary hair cells and possess properties required for the proper structural integration with primary hair cells and supporting cells.

## Induced hair cells demonstrate voltage-dependent ion currents

To determine if iHCs possess electrophysiological properties similar to those of primary hair cells, we performed whole-cell patch-clamp recordings. We measured the biophysical properties of our cells in voltage-clamp and current-clamp to analyze the voltage-gated currents and passive membrane properties of these cells. We compared primary hair cells (n = 5) with *Atoh1*-nGFP+ iHCs in two experimental conditions: monolayer-cultured iHCs (n = 10) and iHCs co-cultured with dissociated organ of Corti (n = 10). Within co-cultures, the presence of the *Atoh1*-nGFP reporter enabled specific patch clamp analysis of iHCs.

Current-clamp was used to measure the passive membrane properties of primary hair cells, co-cultured iHCs and monolayer-cultured iHCs (*Figure 6C*). The properties measured included the resting potential, membrane capacitance, and input resistance (*Figure 6D*). As a negative control we patch-clamped mouse embryonic fibroblasts. The MEFs showed gross electrophysiological properties that did not overlap with that of primary hair cells or induced hair cells (data not shown). The mean resting potentials for primary hair cells, cocultured iHCs, and monolayer-cultured iHCs were −58.6 mV (± 6.9), −54.8 mV (± 4.1), and −50.8 mV (± 2.4), respectively (*Figure 6D*). These values are comparable to previously reported primary hair cell resting potentials (*Dallos, 1985*; *Oliver et al., 2003*). The input resistances were measured to infer the total ion channel composition of the cell. Higher input resistance values indicate the cell may have fewer ion channels to allow current to flow in and out of the plasma membrane. The input resistance was highest in monolayer-cultured iHCs (3878 ± 557 MΩ). However, the input resistance of co-cultured iHCs (1432 ± 327 MΩ) was comparable to that of primary hair cells (1950 ± 755 MΩ) (*Figure 6D*). Lastly, the capacitance, which can be used to infer the surface area of the cell, was highest in primary hair cells (8.4 ± 3.1 pF), followed by co-cultured iHCs (5.6 ± 2.2 pF) and then monolayer-cultured iHCs (4.2 ± 1.2 pF) (*Figure 6D*).

In addition, we performed voltage-clamp to measure the magnitude and time dependent activity of the whole-cell currents in primary hair cells, co-cultured iHCs and monolayer-cultured iHCs (*Figure 6E*). In response to the applied voltage, both primary hair cells and iHCs produced positive-outward currents (*Figure 6E*). However, the monolayer-cultured iHCs produced relatively small whole-cell currents that rapidly inactivated (*Figure 6E*). In contrast, primary hair cells and co-cultured iHCs displayed robust outward currents that more slowly inactivated over the course of the protocol (*Figure 6E*). We measured the steady-state outward current as a function of the voltage-clamp potential and normalized the current magnitude by the cell's capacitance to analyze current densities. Monolayer-cultured iHCs showed small current densities while the co-cultured iHCs and primary hair cells displayed overlapping magnitudes of voltage-dependent current densities (*Figure 6F*).

A prominent voltage-clamp feature in primary hair cells is a delayed onset of a slow-activating outward current (*Housley and Ashmore, 1992*; *Marcotti and Kros, 1999*). In order to measure the kinetic properties of this slow-activating outward current, we fit a single exponential at the onset of the current (*Figure 6G*) to compare the mean time constants when the cells were clamped from −120 mV to 70 mV (*Figure 6H*). The delayed onset current of monolayer-cultured iHCs displayed fast time constants (*Figure 6H*). In contrast, the co-cultured iHCs and primary hair cells showed similarly longer time constants, indicating that their outward currents have similar activation kinetics (*Figure 6H*). Together, these electrophysiological data suggest that both monolayer and co-cultured iHCs possess voltage dependent currents, however, when iHCs are co-cultured with dissociated organ of Corti, their size, passive membrane properties and ion channel function are more similar to those of primary hair cells.

## Induced hair cells possess rudimentary mechanotransduction properties

Primary sensory hair cells acquire distinct functional properties early in development in order to properly convert mechanical sound waves into neurotransmitter signaling (*Wu et al., 2017*). Mechanotransduction relies on the organization of stereocilia, the assembly of tip links, and insertion of mechanically gated ion channels at the tip of each stereocilia (*Kawashima et al., 2011*; *Pan et al., 2013*). Mechanotransduction channels are highly permeable to styryl dyes, and their accumulation in hair cells occurs with much faster kinetics than most other cells (*Gale et al., 2001*).

Primary hair cells within the intact organ of Corti rapidly and selectively accumulate the styryl dye FM4-64 within seconds, a time frame consistent with entry of the dye through mechanotransduction channels rather than endocytosis (*Lelli et al., 2009*; *Figure 6—figure supplement 1D*). In contrast, MEFs failed to incorporate FM4-64 within the 30 s time frame (*Figure 6—figure supplement 1E*). However, iHCs rapidly incorporated FM4-64 to high levels within a 30 s time course (*Figure 6I*) demonstrating that iHCs possess rudimentary mechanotransduction channels with similar styryl dye uptake as in primary hair cells within the intact organ of Corti (*Figure 6—figure supplement 1D*) and primary hair cells from dissociated organ of Corti preparations (*Figure 6J*).

## Induced hair cells recapitulate sensitivity to gentamicin, a known ototoxin

Environmental and pharmacological ototoxins that cause selective degeneration of hair cells are major contributors to hearing loss worldwide (*Al-Malky et al., 2015*; *Sagwa et al., 2015*; *Knight et al., 2017*). Gentamicin is representative of a large class of highly effective aminoglycoside antibiotics that result in significant hair cell degeneration (*Alharazneh et al., 2011*). Unfortunately, a lack of mammalian models suitable for large scale screening of ototoxins and otoprotectants has restricted the development of small molecules to reduce ototoxicity and identification of compounds that can protect against known ototoxins.

To determine if iHCs are sensitive to ototoxic compounds, we tested their ability to accumulate gentamicin in a similar manner to primary hair cells. Primary hair cells of the organ of Corti specifically accumulated Texas-Red conjugated gentamicin (GTTR), but not Texas Red (TR) alone when treated with 0.5 mM of either compound for 3 hr (*Figure 7—figure supplement 1A, B*). MEFs transduced with a GFP-expressing control virus did not accumulate GTTR (*Figure 7A*). The iHCs, similarly to primary hair cells, robustly accumulated gentamicin-Texas Red (GTTR) (*Figure 7B*), but not Texas Red (TR) alone (*Figure 7—figure supplement 1C*), after a 3 hr treatment at 0.5 mM.

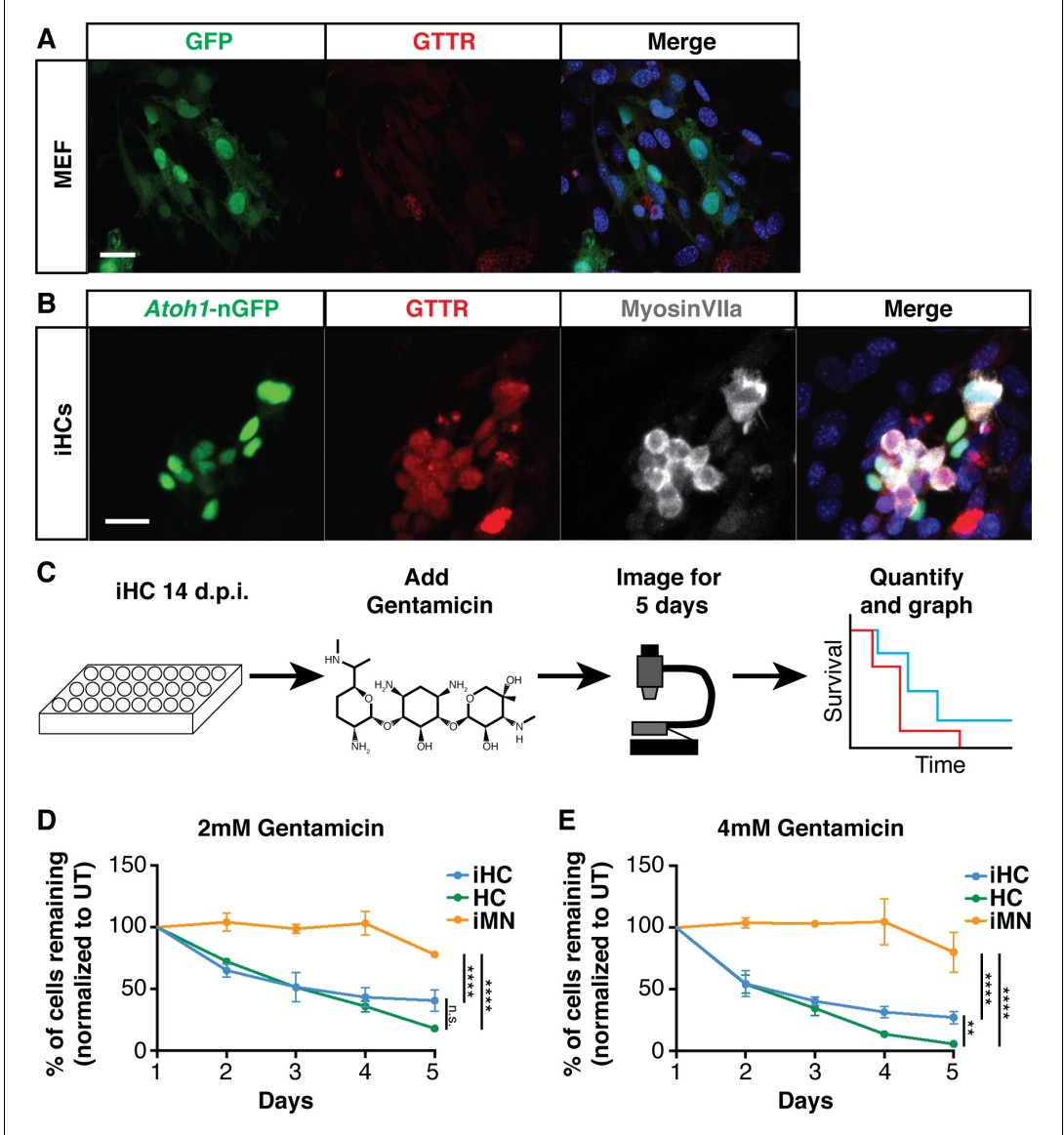

**Figure 7.** Induced hair cells recapitulate susceptibility to a known ototoxin, Gentamicin. (A) MEFs infected with eGFP control virus do not accumulate Gentamicin-Texas Red (GTTR). MEFs were treated with 0.5 mM GTTR for 3 hr. Merged image includes Hoechst nuclear stain. Scale bar represents 50 um. (B) iHCs can accumulate GTTR. iHCs were treated with 0.5 mM GTTR for 3 hr. iHCs were also labeled for anti-MyosinVIIa (grey). Merged image includes Hoechst nuclear stain. Scale bar represents 50 um. (C) Schematic of experimental design for longitudinal survival of *Atoh1*-nGFP+ iHCs. (D–E). Longitudinal survival tracking of P1 hair cells (HC) from dissociated organ of Corti preparations, induced hair cells (iHCs) and induced motor neurons (iMNs) treated with gentamicin at 2 mM and 4 mM respectively. (n = 3 replicates each; mean ± SEM; Two-Way ANOVA *p<0.05, **p<0.01, ***p<0.001, ****p<0.0001).

The online version of this article includes the following figure supplement(s) for figure 7:

**Figure supplement 1.** Induced hair cells recapitulate susceptibility to a known ototoxin, Gentamicin.

To assess whether the gentamicin accumulation seen by GTTR treatment could cause iHCs to degenerate, we established a longitudinal survival assay using robotic imaging and automated tracking of iHC survival (*Figure 7C*). To selectively identify and track survival of *Atoh1*-nGFP+ iHCs from daily whole-well images, we customized a time-lapse nuclei count recipe running on SVCell RS 4.0 (a product of DRVision Technologies that has been rebranded to Aivia). The software can automatically detect and count iHCs based on nuclei morphology and the *Atoh1*-nGFP fluorescence, with comparable results to manual counting (p=0.53). We performed the survival assay with iHCs, dissociated

primary P1 cochlear hair cells (HC) as a positive control, and induced motor neurons (iMNs) as a negative control. While gentamicin caused primary hair cell degeneration in a dose-dependent manner, it caused little-to-no toxicity to *Hb9*-RFP+ induced motor neurons generated from MEFs by transduction with Ngn2, Isl1, Lhx3, Ascl1, Brn2, and Mty1l (*Son et al., 2011*; *Figure 7D,E*). Similar to primary hair cells, iHCs treated with gentamicin showed rapid, dose-dependent degeneration (*Figure 7D,E*). These data indicate that iHCs possess functional properties of primary hair cells and display selective vulnerability to the known ototoxin, gentamicin. Moreover, these results suggest that iHCs provide a scalable platform for detecting agents that protect against gentamicin ototoxicity.

## Discussion

The discovery of treatments for hearing loss, as well as screening for drugs that can protect the sensory hair cells of the inner ear from environmental stress, such as chemotherapy, have been hampered by the small number and inaccessibility of the sensory cells of the inner ear. The current study is an effort to alleviate this problem through the use of direct lineage reprogramming of somatic cells to generate the large numbers of induced hair cells needed for high-throughput screening. To this end, we have succeeded in identifying a cocktail of transcription factors, *Six1, Atoh1, Pou4f3*, and *Gfi1* (SAPG), that is able to reprogram somatic cells to a sensory hair cell-like fate using alternately, mouse embryonic fibroblasts, adult tail tip fibroblasts, and postnatal day eight supporting cells. The reprogramming efficiency using this SAPG cocktail (35%) is similar and in some cases higher than the best reprogramming strategies for other cell types such as cardiomyocytes with 20% efficiency (*Ieda et al., 2010*) and motor neurons with 10–40% efficiency (*Ichida et al., 2009*; *Babos et al., 2019*). With this method, we show that the reprogrammed hair cells are transcriptionally and epigenetically similar to perinatal primary cochlear hair cells, including morphology, physiology, and susceptibility to ototoxic agents, specifically gentamicin. We also demonstrate a method for high-throughput screening, that in the future will allow the discovery of new otoprotectants, as well as gene-therapy/regenerative medicine approaches to treat hearing loss.

### Reprogramming efficiency and the level of maturity of iHCs

As confirmation of the importance of the four transcription factors identified in our unbiased reprogramming investigation, three of the four transcription factors, *Atoh1, Pou4f3*, and *Gfi1*, were previously used to induce a hair cell like-fate from mouse ES cells that had been partially differentiated into ectodermal organoids (*Costa et al., 2015*). In our hands, these factors are able to activate reporter expression in MEFs from *Atoh1*-nGFP mice, but yield a mixed population in which many GFP-positive cells failed to upregulate the hair cell markers MyosinVIIa and Parvalbumin. We hypothesize that the addition of *Six1* increases reprogramming efficiency by pushing cells towards the sensory ectodermal lineage. *Six1* has been reported previously to promote competency and progenitor-like state, as well as being an essential determinant of early sensory inner ear lineage (*Zheng et al., 2003*; *Ozaki et al., 2004*; *Zhang et al., 2017*). The expression gradient of *Six1* in the developing sensory epithelium precedes the activation of *Atoh1* (*Ahmed et al., 2012*). *Six1* directly targets the *Atoh1* autoregulatory enhancer, as well as other essential hair cell enhancers for *Pou4f3* and *Gfi1*, with an increase in *Six1* occupancy as hair cells differentiate with in the sensory epithelium (*Li et al., 2020*). Additionally, *Six1* has been shown to play a role in the maturation of hair cells by regulating key genes involved in the establishment of planar cell polarity and hair-bundle orientation (*Li et al., 2020*). These studies support our findings that, in addition to *Atoh1, Pou4f3* and *Gfi1, Six1* is an important upstream transcription factor in establishing a hair cell-specific gene expression program in our direct reprogramming.

At perinatal times, only modest differences in gene expression are able to distinguish inner and outer hair cells of the cochlea (*Liu et al., 2014b*), as well as vestibular vs. cochlear hair cells (*Burns et al., 2015*). The transcriptional profiles of postnatal day one cochlear hair cells and utricular hair cells are very similar (*Figure 2—figure supplement 1A*). This time point represents an immature hair cell, and the transcriptional profiles are known to change as the hair cells mature and acquire subtype specificity and functionality (*Burns et al., 2015*; *Zhu et al., 2019*; *Hoa et al., 2020*). As a result of comparing iHCs to postnatal day one primary hair cells, we are unable to statistically classify iHCs as being more similar to one or another hair cell type based on the bulk RNA sequencing.

While many of the specialized gene characteristics of the cochlear hair cells are clearly upregulated during reprogramming (*Figure 2*), the iHCs fail to activate other important genes essential for the functional maturation of the sensory receptors in the cochlea, such as *Prestin* and *Gata3* (*Liberman et al., 2002*; *Bardhan et al., 2019*).

The imperfections of iHC reprogramming could have several causes. First, as noted, we could be lacking additional, hair cell-type transcription factors to drive cells to a more mature state. For example, the transcription factor Zfp503 identified in the initial set of 16 factors for reprogramming, does not get activated in response to the SAPG reprogramming cocktail. We saw that the addition of Zfp503 to the core group SAPG negatively impacted the efficiency of activation of the *Atoh1*-nGFP reporter (*Figure 1—figure supplement 1G*), however it remains to be investigated whether the addition of Zfp503 can confer a more mature induced hair cell phenotype. Zfp503 is highly expressed in postnatal day one cochlear hair cells, but not in utricular hair cells, suggesting it could play an important role in conferring subtype specificity in the iHC reprogramming. Additional bioinformatic analysis has shown that some of the relatively small group of distal regulatory elements that are present in P1 hair cells, but fail to open in iHCs, associate with hair cell-specific genes that also fail to be robustly expressed during reprogramming (data not shown). This further suggests additional factors, perhaps such as Gata3, that may improve the quality and maturity of the iHCs. In addition, transcriptional characterization of older hair cells will allow for identification of additional TFs, which may improve the reprogramming strategy. Second, our current strategy relies on constitutive expression of the reprogramming factors, while continuous expression of *Atoh1* is known to halt hair cell maturation (*Liu et al., 2012b*; *Liu et al., 2014a*). We plan to overcome this limitation by using inducible gene expression constructs to drive reprogramming in the future. Finally, lack of organ-specific context in vitro may not provide additional signals for maturation. In fact, we demonstrate that iHCs co-cultured with dissociated organ of Corti cells, promoted morphological and electrophysiological maturation of iHCs (*Figure 6*). This functional maturation may be mediated, at least in part, by improved trafficking and assembly of ion channel subunits conferred by the co-cultures. We plan to examine the transcriptional profile changes that occur in iHCs after co-culture in order to understand what genes may be contributing to the morphological and functional maturation.

As has been documented in other reprogramming experiments (*Wapinski et al., 2013*; *Wapinski et al., 2017*; *Rhee et al., 2017*), the chromatin landscape was also drastically remodeled during reprogramming of MEFs to a hair cell-like state, readily opening de novo distal elements to change their chromatin to resemble P1 cochlear hair cells. Similar to the RNA sequencing results, there is a residual MEF signature of the chromatin landscape, and the failure to close down these chromatin regions may be acting as a barrier for more efficient and/or faithful reprogramming. In addition, the opening of a large number of peaks inappropriately (peaks that occur in neither MEFs of primary P1 hair cells) suggests that our cocktail may lack a transcriptional repressor, and that these inappropriately open distal elements may provide some explanation for the low level of inappropriate gene expression (*Figures 2* and *4*).

Despite these pitfalls, common to most if not all reprogramming strategies published to date (*Ieda et al., 2010*; *Son et al., 2011*; *Treutlein et al., 2016*; *Kaminski et al., 2016*; *Van Pham et al., 2017*), induced hair cells are highly similar to their primary counterparts on transcriptional and epigenetic levels, as well as functional levels.

## Direct reprogramming as a strategy for gene therapy and identification of genetic causes of hearing loss

Recently the use of Anc80-based adeno associated vectors made inner ear gene delivery feasible (*Suzuki et al., 2017*; *Tao et al., 2018*; *Tan et al., 2019*). The proof of concept studies have demonstrated functional recovery after administration of TMC1 gene therapy in animals carrying a mutation in the gene (*Yoshimura et al., 2019*). Yet, the genetic causes of deafness are often unknown in patients, and loss of hair cells remains the leading contributor to hearing loss worldwide. We found that the SAPG four transcription factor combination is significantly more effective at activating the expression of hair cell genes MyosinVIIa and Parvalbumin in adult tail tip fibroblasts, and postnatal (P8) supporting cells, when compared to *Atoh1* alone (*Figure 5*). Supporting cells have been shown to be maintained in long deafened mice (*Oesterle and Campbell, 2009*) and humans (*Johnsson et al., 1981*). The demonstration that SAPG is able to convert P8 supporting cells to a hair cell-like fate, highlights the potential for future gene therapy approaches for hearing loss.

The ability to reprogram cells to a hair cell fate provides new opportunities to target hearing loss through the development of disease-specific drug screens and personalized medicine. Primary human fibroblasts taken from patients, can be reprogrammed to induced pluripotent stem cells that can then be expanded and reprogrammed to a hair cell fate for patient- and disease-specific studies, opening the opportunity to study hearing loss mutations, ototoxicity and regenerative medicine approaches (*Koch et al., 2011*; *Lim et al., 2016a*; *Lim et al., 2016b*; *Shi et al., 2018*; *Huang et al., 2019*; *Villanueva-Paz et al., 2019*; *Lee et al., 2019*).

## Reprogramming strategy vs other approaches

The evaluation of ototoxic compounds, and the identification of new otoprotectants has been severely limited by the lack of sufficient numbers of mammalian hair cells available for study. Although directed differentiation of ESCs towards the sensory hair cells have been described (*Li et al., 2003*; *Oshima et al., 2010*; *Koehler et al., 2013*; *Ronaghi et al., 2014*), these three-dimensional protocols are time consuming and the outcomes are ESC-line dependent with variable efficiency across protocols (*Hiler et al., 2015*; *Mellough et al., 2019*; *Yoon et al., 2019*). Direct reprogramming allows for a much faster and more reliable approach to produce large quantities of induced hair cell-like cells to scale, and the monolayer culture used here overcomes several of the short-comings of directed differentiation systems presented so far, such as long culture periods (*Koehler et al., 2013*), and 3D cultures that are more difficult to image robotically (*Breslin and O'Driscoll, 2013*; *van Vliet et al., 2014*). The efficiency and reproducibility of direct reprogramming is essential for high throughput screening approaches. We demonstrated that iHCs are selectively vulnerable to gentamicin, and can be reprogrammed and cultured in microtiter plate format, providing a robotic-imaging platform for scalable monitoring of iHC survival and small-molecule screening. Importantly, reprogramming does not require an iPSC intermediate, thus generating iHCs from human patients with known or novel genetic mutations associated with hearing loss will enable screening for new therapeutic targets and agents for the treatment of genetic causes of deafness.

The direct lineage conversion of somatic cells to a hair cell-like fate provides the means to study many outstanding questions in inner ear biology. For instance, the nature of gene regulatory logic underlying the differentiation of the variety of cochlear and vestibular hair cell types, as well as mechanisms underlying hair cell degeneration caused by ototoxins, and the numerous mutations responsible for the many types of syndromic and non-syndromic hearing loss.

## Materials and methods

### Contacts for reagent and resource sharing

Further information and requests for resources and reagents used in this study should be directed to the Lead Contacts, Dr. Justin Ichida (ichida@usc.edu) or Dr. Neil Segil (nsegil@med.usc.edu).

### Mice

All experiments were performed at the University of Southern California. All animal experiments were conducted according to the National Institutes of Health Guide for Care and Use of Laboratory Animals. Protocols and experiments using animals were approved by the Institutional Animal Care and Use Committee at the University of Southern California.

Mice were housed with free access to chow and water and a 12 hr day/night cycle. Breeding and genotyping of the mice was performed according to USC standard procedures.

*Atoh1*-nGFP (previously known as Math1-GFP) transgenic line was obtained from Jane Johnson. *Atoh1*-nGFP transgenic mice were mated with wild type CD1 mice to obtain litters for mouse embryonic fibroblast isolations and tail tip fibroblast isolations. Wild type CD1 mice were used for harvesting wild type organs of Corti in co-culture experiments. Lfng-CreERt2::Rosa26$^{tdTomato}$ transgenic mice were used for harvesting organs of Corti with lineage traced supporting cells.

### Molecular cloning of viral plasmids and virus production

Complimentary DNAs for the 16 candidate transcription factors were each cloned into viral expression vectors using the Gateway cloning (Invitrogen). Retroviral and lentiviral plasmids were

constructed into the entry vector pDONR221. Entry clones were recombined into destination vectors via LR reaction into the pMXS-DEST (retro) or FUWO-tetO-DEST (lenti).

Plat-E cells and HEK293 cells were cultured in MEF medium (DMEM containing 10% fetal bovine serum) and used to produce retroviruses and lentiviruses respectively. Cells were transfected at 90% confluency with viral vectors containing genes of interest and viral packaging plasmids (PIK-MLV-gp and pHDM for retrovirus; pPAX2 and VSVG for lentivirus) using linear polyethylenimine (PEI) (Sigma-Aldrich). After 24 hr of incubation with the plasmid DNA and PEI, the medium was replaced with fresh MEF medium and the culture was continued. Supernatants from the transfected cells were collected at 24 hr and 48 hr after medium replacement, filtered through 0.45 um filters and used immediately if generated from Plat-E or concentrated using Lenti-X concentrator (Clontech) and stored at −80°C if generated from HEK293T.

MEFs were transduced by mixing virus with MEF media. The virus containing media was removed from the MEFs after 24 hr and replaced with MEF media. The next day, medium was switched to hair cell medium (HCM: DMEM/F-12 (supplier), N2 and B27 supplements (supplier), EGF (2.5 ng/ml), and FGF (5 ng/ml)).

## MEF isolation

Mouse embryonic fibroblasts (MEFs) were obtained from E13-14 embryos taking care to exclude contamination with other Atoh1 expressing tissues (kidney, brain, spinal cord and webbing between digits). Tissue was minced with a razor blade and enzymatically dissociated with 0.25% trypsin-EDTA for 30 min at 37°C. Trypsinization was quenched by addition of MEF media (previously described). The isolated cells were centrifuged (800 g for 10 min) and the pellet was resuspended in MEF media before plating onto gelatin coated T75 tissue culture flasks. We found that plating two embryos per T75 gave optimal survival post-dissection. The MEFs were cultured until confluency was reached and then cryopreserved in liquid nitrogen using freezing media (1:1 mixture of MEF media and Freezing media (FM; 80% fetal bovine serum and 20% DMSO)). MEFs were used without further passaging for reprogramming experiments. All cells were tested for mycoplasma contamination and came back negative.

## TTF isolation

Tail tip fibroblasts (TTFs) were obtained from 6 month old adult Atoh1-nGFP transgenic mice. The tail was harvested from the sacrificed mouse by removing the skin and dissecting the remaining tail tissue into small segments. After plating on gelatin coated dishes, the tissue adhered to the dish and the expanding cells eventually covered the dish. The TTFs were harvested for reprogramming by simply moving the segments to a new dish and then collecting the remaining adherent cells. TTFs were cultured in DMEM with 40% fetal bovine serum. TTFs were frozen in liquid nitrogen and used without further passaging for reprogramming experiments. All cells were tested for mycoplasma contamination and came back negative.

## Merkel cell isolation

Skin was obtained from postnatal day 1 (P1) mice. Skin was incubated overnight at 4°C in Accutase. The epithelium (epidermis and hair follicles) was separated from the underlying dermis with forceps and the epidermal cells were dissociated with trypsin for 10 min at 37°C then dissociated to single cell suspension. The freshly isolated epidermal cell suspension was then FACS purified to sort for *Atoh1*-nGFP positive Merkel cells.

## Gut secretory cell isolation

Small intestines were obtained from adult mice. Intestinal villi were scraped away, crypt epithelium was collected by shaking in 5 mM EDTA for 50 min at four degrees Celsius, and single cell suspensions were prepared by digestion in 4x TrypLE (Invitrogen) for 50 min at 37°C. The freshly isolated cell suspension was then FACS purified to sort for *Atoh1*-nGFP positive cells gut secretory cells.

## Cerebellar granule precursor isolation

Cerebellums were obtained from postnatal day 1 (P1) mice. Tissue was minced and enzymatically digested using 0.25% trypsin for 10 min at 37°C then dissociated to single cell suspension. The

freshly isolated cerebellar cell suspension was then FACS purified to sort for *Atoh1*-nGFP positive cells cerebellar granule precursors.

## Primary hair cell culture

The primary hair cell culture was established by dissecting the organs of Corti from P1 transgenic *Atoh1*-nGFP mice. The cells were dissociated to a single cell suspension and plated onto laminin coated tissue culture plates or cover slips.

The primary cultures from Lfng-CreERt2::Rosa26^tdTomato transgenic mice were done at postnatal day 8 (P8). Lfng-CreERt2::Rosa26^tdTomato transgenic mice were injected with tamoxifen at postnatal day three for lineage tracing of the Lfng+ supporting cell population. The organs of Corti were harvested at P8, dissociated to single cell suspension in HCM and the reprogramming factors were added to the cell suspension. The cells were then plated onto laminin coated tissue culture treated cover slips. The virus containing media was replaced after 24 hr with fresh HCM.

All primary cultures were plated using ROCK Inhibitor (Y-27632) (Sigma-Aldrich) for the first 24 hr to help promote survival.

## Co-culture of iHCs

Induced hair cells were FACS purified to obtain the *Atoh1*-nGFP positive cells and collected in HCM. The primary organ of Corti was dissected from wild type mice at E13.5 and enzymatically dissociated to a single cell suspension containing primary hair cells, primary supporting cells and a portion of the surrounding periotic mesenchmye. The iHCs were then mixed with the dissociated organs of Corti. The ratio of iHC to cells of the organ of Corti was kept at about 1:33. This ratio was determined from the fact that the primary organ of Corti contains approximately 3000 hair cells and upon dissociation gives approximately 100,000 total cells. Co-cultures were grown on tissue culture treated coverslips in wells of 24 well plates that had been coated with a 20 ul drop of matrigel (10% in HCM) at the center of the coverslip. The co-culture cell suspension was plated as 30 ul drops (2,500–3,000 cells per ul) in the center of the matrigel coated drop on the cover slip. 12–24 hr after plating the drops the well was flooded with 500 ul of HCM. All cocultures were maintained in HCM.

## Immunostaining

Cells for staining were washed with PBS and fixed using 4% paraformaldehyde (PFA) in phosphate-buffered saline (PBS) for 15 min at room temperature. For permeabilization and blocking the cells were incubated in PBST (0.1% Triton-X 100 in PBS) with 10% fetal bovine serum for 2 hr at room temperature or overnight at 4˚C. After blocking, the cells were washed 3 times for 5 min with PBS. Cells were then incubated with the primary antibody overnight at 4˚C. Then the cells were washed with PBS again before incubation with the secondary antibody for one hour at room temperature or overnight at 4˚C. Primary and secondary antibodies were diluted in PBST with 10% serum. The DNA was stained with Hoechst diluted 1:1000 in PBS for 10 min at room temperature.

## Antibodies

    Anti-Parvalbumin (Sigma-Aldrich, catalog# P3088-100UL)
    Anti-Espin (Gift from Hudspeth Lab)
    Anti-MyosinVIIa (Proteus Bioscience, catalog# 25–6790)
    Molecular Probes Phalloidin Rhodamine (Thermo Fisher Scientific, catalog# R415)
    Phalloidin-iFluor 647 Reagent (Abcam Biochemicals, catalog# ab176759)
    Anti-acetylated Tubulin (Sigma Aldrich, catalog# T7451-100UL)
    Anti-Sox2 (Abcam, catalog# ab97959)

## Imaging

Immunostaining images of adherent cell cultures were acquired on an LSM780 confocal microscope using Carl Zeiss Zen blue/black software and processed using Adobe Illustrator CS6 software. For quantification of reprogramming efficiency in adherent cultures, images were acquired at 10x using the Molecular Devices ImageExpress. The images were either processed manually using ImageJ software and the Cell Counter plug in or automatically using SVCell RS (described below). Counts are

represented as reprogramming efficiency (percent of *Atoh1*-nGFP+ cells per well of 5000 MEFs infected).

## iHC detection and counting method

Automated cell counting used thresholds for size, intensity and roundness of the *Atoh1*-nGFP signal. The imaging was done at 10x. For each time frame, the customized time-lapse nuclei count recipe of SVCell RS is applied to first reduce noise with image smoothing. Objects are detected by performing background removal followed by adaptive thresholding. A size filtering is then applied to remove objects that are either too large or too small. The count of remaining objects is measured for the time point. Batch processing is available for applying the recipe to multiple time-lapse images and saving results. To ensure the reliability of the automated counting a comparison was done of 20 wells counted both manually and automatically (p=0.53).

## Flow cytometry preparations

Primary hair cells were harvested from *Atoh1*-nGFP transgenic mice. The cochleas were incubated in 0.25% trypsin for 8 min and gently triturated to single cell suspension. Media (DMEM with 10% FBS) was added to the dissociated cells and then spun down at 1000 rpm for 5 min, resuspended in Hair Cell Media, passed through a 70 um cell strainer and then FACS-purified). The same procedure was used to FACS-purify dsRED MEFs and *Atoh1*-nGFP+ cells from the reprogramming cultures.

## RNA sequencing

Total RNA was extracted from primary mouse hair cells (at postnatal day 1), mouse iHCs (at day 14 post infection with reprogramming factors) and MEFs (at 14 days post transduction with dsRed). For each replicate 20,000 FACS-sorted cells were used as input for RNA-seq. Total RNA was extracted with either Quick-RNA Microprep kit (Zymo Research), quantified by bioanalyzer and then processed for libraries with either QIAseq FX Single Cell RNA Library Kit (Qiagen) or TruSeq RNA Library Prep Kit v2 (Illumina). Specific sequencing parameters and instrument models were submitted with GEO datasets. At least three replicates were collected for each condition and sequenced to a depth of at least 20 million reads.

Reads were mapped to the mouse reference genome (Gencode Mm10v11) using STAR. Read counts were quantified by RSEM. Only protein coding polyA tail transcripts and autosomal genes were kept. Transcript counts were collapsed to gene counts. Differentially expressed genes were identified using the DESeq2 package. Genes with a log fold change threshold greater than one and adjusted P-value of less than 0.1 were considered significant. Principle component analysis and unsupervised hierarchical clustering of RNA-seq was performed using counts transformed by DESeq2's regularized logarithm (Rlog).

GEO accession number: GSE149260.

## GO analysis

Gene ontology analysis was performed on categorized gene sets using R clusterProfiler package. GO results were visualized using the R enrichplot package.

## GSEA analysis

Gene Set Enrichment Analysis was performed using the R package fgsea. The Wald statistic from the differential comparison of reprogrammed cells versus MEFs was used to pre-rank genes for subsequent GSEA analysis. Gene sets representing unique signatures for each Atoh1 positive cell-type were tested for enrichment in SAPG. To determine signature gene sets for each Atoh1 cell type, only genes with a log2 foldchange greater than or equal to two with adjusted P-value less than 0.01 compared between each profiled Atoh1 positive cell-type were used. Utricle and cochlear hair cells were treated as a single cell type due to small number of unique genes at the postnatal day one developmental stage used.

## ATAC-seq

Cells were collected by FACS purification into cold PBS, and centrifuged 500 xg for 15 min. Cell pellet were resuspended with 50 ul transposition buffer consisting of 10 mM Tris-HCL pH8.0, 5 mM

MgCl2, 10% DMF, 0.2% NP40, and home-made transposase Tn5. Transposition was performed at 37°C for 20 min. DNA was collected immediately after transposition using Qiagen Mini-elute kit.

Encode pipeline was adapted for alignment and QC for ATAC-seq and ChIP-seq data. Paired-end reads were quality trimmed with cutadapt (v1.18) and aligned to mouse reference genome (Gencode Mm10v11) with bowtie2 (v2.2.6) using parameters -X2000 -mm –local. PCR duplicates were removed based on genomic coordinates. Only autosomal chromosomes were selected and used for downstream analysis.

Specific sequencing parameters and instrument models were submitted with GEO datasets.

## ChIP-seq

Histone ChIP-seq protocol was developed by us based on µChIP-PCR protocol published previously (Stojanova et al., 2016) with additional Tn5 tagmentation step. Briefly, chromatin was cross-linked with 1% formaldehyde (Thermo Fisher) for 8 min, quenched with 125 mM Glycine (Sigma) for 5 min at room temperature, sonicated using the microtip of a High Intensity Ultrasonic Processor (Sonics and Materials, Newtown, CT; amplitude 50, power 50) for $8 \times 30$ s with 30 s pause, tagmentated with Tn5 transposase for 30 min at 37°C, incubated with antibody complexed with Dynabeads Protein A (Thermo Fisher) overnight at 4°C, precipitated and washed three times on magnetic rack, and finally PCR amplified with primers matching Tn5 adapters.

Encode pipeline was adapted for alignment and QC for ChIP-seq data. Paired-end reads were quality trimmed to 36 bp with cutadapt (v1.18) and aligned to mm10 reference genome (Gencode Mm10v11) with STAR aligner using parameters end-to-end and alignIntronMax = 1 for DNA alignment. PCR duplicates were removed with STAR. Only autosomal chromosomes were selected and used for downstream analysis.

Specific sequencing parameters and instrument models were submitted with GEO datasets.

## Chromatin analysis

ATAC peaks and H3K27ac peaks were identified using the R package chromstaR (parameters: binsize = 500 bp, stepsize = 250 bp, mode = full). An equal number of reads were randomly sampled for H3K27ac replicates (17.5 million) and ATAC replicates (15 million reads) as input for subsequent chromatin analysis. For peak calling, a false discovery rate (FDR) cutoff of 0.01 and 0.001 was used for ATAC and H3K27ac respectively and an RPKM cut off >2. Promoter regions were defined by 2 kb upstream of 500 bp downstream of protein coding transcription start sites; all remaining regions were considered distal. Enhancers were defined by cooccurrence of an ATAC peak and H3K27ac peak at distal regions.

Differential ATAC peak analysis was performed between P1HC, SAPG iHCs, and MEFs using chromstaR. Regions with a differential score of at least 0.999999 were considered differentially accessible. Regions with differential score less than 1E-06 were considered non-differentially accessible.

Deeptools was used to average replicates and calculate coverage tracks and for ATAC-seq and ChIP-seq data for visualization on IGV and heatmaps.

## Electrophysiology

Whole cell patch clamping was performed on three different preparations of cells. The first preparation was iHCs in the monolayer culture of MEFs at D14-15 post infection with SAPG. The second preparation was iHCs FACS purified and replated with dissociated wild type organ of Corti. The third preparation was postnatal day one primary hair cells from the dissociated transgenic *Atoh1*-nGFP organ of Corti. Preparations were viewed at X630 using a Zeiss Axios Examiner D1 microscope fitted with Zeiss W Plan-Aprochromat optics. Signals were driven, recorded, and amplified with an Multiclamp 700B amplifier, Digidata 1440 board and pClamp 10.7 software (pClamp, RRID:SCR_011323). Recording and cleaning pipettes were fabricated using filamented borosilicate glass. Pipettes were fired polished to yield an access resistance between 4–8 MΩ. Each recording pipette was covered in a layer of parafilm to reduce pipette capacitance. Recording pipettes were filled with standard internal solution. The contents of the standard internal solution are (in mM): 135 KCl, 3.5 MgCl2, 3 Na2ATP, 5 HEPES, 5 EGTA, 0.1 CaCl2, 0.1 Li-GTP, and titrated with 1M KOH to a pH of 7.35 and an osmolarity of about 300 mmol/kg. The voltage clamp protocol was performed by

holding the cell at −60 mV followed by a stimulus of voltage steps (−120 to +70 mV, by intervals of 10 mV). The current response of the cell was recorded along with measures of ionic current peak amplitudes, channel conductance values, and current activation kinetics.

Analysis of the data was performed using a combination of pClamp (pClamp, RRID:SCR_011323), Matlab (MATLAB, RRID:SCR_001622), JMP (JMP, RRID:SCR_014242), Origin Pro (OriginPro, RRID: SCR_015636), and Imaris (Imaris, RRID:SCR_007370). pClamp software was be used to gather and quantify raw data from electrophysiological recordings.

## FM lipophilic styryl dye uptake assay

Cells were incubated with 1 uM FM 4-64FX, the fixable analog of FM4-64 (Life Technologies, catalog# F34653). Prior to incubation the FM 4-64FX was resuspended in ice cold HBSS at a 1 mM concentration. The cells were incubated with a final concentration of 1 uM FM 4-64FX in ice cold HBSS for 30 s. After incubation, the cells were rinsed in HBSS three times and then immediately imaged. Using the software ImageJ, the images were filtered on minimum background intensity in order to reduce the amount of background signal. The filter measures the minimum signal intensity found in the image and applies the filter to remove the minimum signal across the entire image. This image enhancement was used uniformly on all images and all channels for each cell type.

## GTTR uptake assay

Gentamicin sulfate salt (Sigma Aldrich catalog# G3632-5G, 50 mg/ml in K2CO3, pH0) and Texas-Red (Thermo Fisher Scientific catalog# T20175, 2 mg/ml in dimethyl formamide) were agitated together overnight to produce gentamicin-Texas Red conjugate (GTTR). The mixture contained 4.4mls of 50 mg/ml gentamicin (GT) with 0.6mls of 2 mg/ml Texas Red (TR) to produce approximately 300:1 molar ratio of GT:GTTR. A high ratio of gentamicin ensures a minimum of unbound Texas Red molecules. The molecular weight of GT is 477.6 g/mol and the molecular weight of TR is 816.94 g/mol. The GTTR was made at a stock concentration of 100 mM. The cells were incubated with HCM containing 0.5 mM or 1 mM GTTR for three hours. After incubation the cells were washed three times with PBS and then immediately fixed using 4% PFA in PBS for 15 min at room temperature.

## Ototoxicity assay

The cells were cultured (for primary hair cells) or reprogrammed (for iHCs and iMNs) in a 96 well tissue culture plate. The primary hair cells were used 24 hr post dissociation of the organ of Corti and plating. The iHCs and iMNs were reprogrammed for 14 days prior to starting the survival assay. The gentamicin was dissolved in HCM at a concentration of 100 mM and subsequently diluted to 8 mM in HCM. The stock at 8 mM in HCM was used for serial dilution to get the desired range of concentrations (8 mM, 4 mM, 2 mM, 1 mM, 0.5 mM, 0.25 mM, and 0.125 mM). The control wells received only HCM.

The assay was performed over a period of 5 days. The HCM containing with or without gentamicin was added to the cells on Day one and the cells were imaged every 24 hr after the initial treatment. Subsequent media changes were performed every other day (Day 3). The HCM for gentamicin treated wells and control wells was made fresh for each media change. The assay ended on Day 5. The Molecular Devices ImageExpress was used for imaging the plate robotically every 24 hr. The images were taken at 10x.

## Statistics

Sample numbers, experimental repeats and statistical test used are indicated in figure legends. Unless otherwise stated, data presented as mean + SEM of at least three biological replicates. Significance summary: $p>0.05$ (ns), $*p\leq0.05$, $**p\leq0.01$, $***p\leq0.001$, and $****p\leq0.0001$.

## Acknowledgements

For technical assistance, we thank Welly Makmura for mouse colony and laboratory supply maintenance, Bernadette Masinsin for flow cytometry support, Mickey Huang and the Choi Family

Therapeutic Screening for high-content imaging and data transfer of our survival assay, and Seth Ruffins from the imaging core for confocal training.

For materials, we thank the Hudspeth Lab for sharing their Espin antibody.

For manuscript revisions, we thank Ksenia Gnedeva (USC) and Andy Groves (Baylor College of Medicine) for critically reading the manuscript and providing constructive feedback and suggestions to improve the clarity and organization of the manuscript.

## Additional information

### Competing interests

Suhasni Gopalakrishnan: Currently employed at a for-profit corporation, Allogene. This company has no competing technology, nor is it involved in this research. Chichou Huang, James Lee: Employed by a for-profit company that will benefit from the software development that they contributed to this project. The other authors declare that no competing interests exist.

### Funding

| Funder | Grant reference number | Author |
|---|---|---|
| National Institutes of Health | R01DC015530 | Justin Ichida<br>Neil Segil |
| Keck School of Medicine of USC | | Justin Ichida<br>Neil Segil |
| National Institutes of Health | R00NS077435 | Justin Ichida |
| National Institutes of Health | R01NS097850 | Justin Ichida |
| New York Stem Cell Foundation | New York Stem Cell Foundation-Robertson Investigator | Justin Ichida |
| Merkin Family Foundation | Richard N. Merkin Assistant Professor of Stem Cell Biology and Regenerative Medicine at USC | Justin Ichida |
| Tau Consortium Investigator Grant | Tau Consortium | Justin Ichida |

The funders had no role in study design, data collection and interpretation, or the decision to submit the work for publication.

### Author contributions

Louise Menendez, Conceptualization, Formal analysis, Investigation, Methodology, Writing - original draft; Talon Trecek, Conceptualization, Resources, Data curation, Software, Formal analysis, Supervision, Funding acquisition, Investigation, Methodology, Project administration, Writing - review and editing; Suhasni Gopalakrishnan, Conceptualization, Formal analysis, Investigation, Methodology, Writing - review and editing; Litao Tao, Conceptualization, Resources, Data curation, Software, Formal analysis, Supervision, Funding acquisition, Validation, Investigation, Methodology, Project administration, Writing - review and editing; Alexander L Markowitz, Formal analysis, Investigation, Methodology; Haoze V Yu, Data curation, Software, Formal analysis, Validation, Investigation, Methodology; Xizi Wang, Justin Ichida, Neil Segil, Conceptualization, Resources, Formal analysis, Supervision, Funding acquisition, Investigation, Methodology, Project administration, Writing - review and editing; Juan Llamas, Data curation, Formal analysis, Validation, Investigation, Methodology; Chichou Huang, Resources, Data curation, Software, Formal analysis, Supervision, Investigation, Methodology, Writing - review and editing; James Lee, Data curation, Software, Formal analysis, Investigation, Methodology, Writing - review and editing; Radha Kalluri, Conceptualization, Resources, Data curation, Formal analysis, Supervision, Funding acquisition, Investigation, Methodology, Project administration, Writing - review and editing

Author ORCIDs
Louise Menendez  https://orcid.org/0000-0001-6830-0358
Radha Kalluri  http://orcid.org/0000-0002-0360-8965
Neil Segil  https://orcid.org/0000-0002-0441-2067

Ethics
Animal experimentation: All experiments were performed at the University of Southern California. All animal experiments were conducted according to the National Institutes of Health Guide for Care and Use of Laboratory Animals. Protocols and experiments using animals were approved by the Institutional Animal Care and Use Committee at the University of Southern California (#20549).

Decision letter and Author response
Decision letter https://doi.org/10.7554/eLife.55249.sa1
Author response https://doi.org/10.7554/eLife.55249.sa2

## Additional files

### Supplementary files
• Transparent reporting form

### Data availability
Sequencing data have been deposited in GEO (accession number GSE149260). Sequence data associated with this paper can be visualized on the gEAR website (https://umgear.org/p?l=e2d98834).

The following dataset was generated:

| Author(s) | Year | Dataset title | Dataset URL | Database and Identifier |
|---|---|---|---|---|
| Menendez L, Trecek T, Gopalakrishnan S, Tao T, Markowitz AL, Yu HZ, Wang XE, Llamas J, Huang C, Lee J, Kalluri R, Ichida J, Segil N | 2020 | Generation of Inner Ear Hair Cells by Direct Lineage Conversion of Primary Somatic Cells | https://www.ncbi.nlm.nih.gov/geo/query/acc.cgi?acc=GSE149260 | NCBI Gene Expression Omnibus, GSE149260 |

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
