## [Decision Letter]

Thank you for submitting your article "Generation of Inner Ear Hair Cells by Direct Lineage Conversion of Primary Somatic Cells" for consideration by *eLife*. Your article has been reviewed by three peer reviewers, one of whom is a member of our Board of Reviewing Editors, and the evaluation has been overseen by Kathryn Cheah as the Senior Editor. The reviewers have opted to remain anonymous.

The reviewers have discussed the reviews with one another and the Reviewing Editor has drafted this decision to help you prepare a revised submission.

Summary:

This exciting study identifies a combination of 4 transcription factors (*Six1*, *Atoh1*, *Pou4f3* and *Gfi1*) to directly reprogram several different types of somatic cells into hair cell-like cells (iHCs). Such in vitro generation of Hair Cells allows screening for otoprotective and regenerative strategies. Previous studies showed that hair cells can be generated from murine pluripotent stem cells either by directed differentiation, human embryonic stem cells or by a combination of directed differentiation to a placodal cell type followed by transcription factor induction. The disadvantage of these techniques is that they require 3D culture conditions which makes screening challenging. The most impressive feature of the paper is a series of assessments of the iHCs compared to the normal HCs. The transcriptomic and epigenetic profiles of iHCs were well characterized, which distinguishes this work from other, less thorough, studies on new har cell induction in vitro. In addition to the fact that the strategy developed by the authors has a high reprogramming efficiency and is already useful for screening purposes the authors frankly discuss any shortcomings of this reprogramming strategy and how they could be overcome. Overall, the manuscript is well-written and logically conveys the dataset. Below are a list of concerns that should be addressed through discussion or experimentation prior to publication:

Essential revisions:

The following comments should be addressed by further experimentation or analysis:

1) It should be evaluated, based on published data, whether the 16 transcription factors initially identified are also expressed by vestibular hair cells. A comparison of iHCs with cochlear hair cells is appropriate based on the rationale presented in the study, but further comparison with vestibular hair cells might reveal an even higher level of similarity.

2) FM dye uptake assay: The images in Figure 6C are not entirely convincing. For instance, it would aid with the interpretation to see FM uptake in a native hair cell vs. iHCs in this co-culture setup.

3) The signature genes for Merkel Cells omit some classic markers, such as *Sox2*, Isl1, Krt20. Is it possible that this dataset contains a mixture of keratinocytes and Merkel cells?

The following comments should be discussed:

1) A powerful experiment that would suggest translatability would be to add the cocktail to rodent utricle cultures treated with an aminoglycoside. Have the authors already attempted this experiment? If yes, the results should be discussed.

2) Postnatal day 8 (P8) cochlear supporting cells: The author explain that supporting cells became an interesting target to drive HC differentiation, but they do not clearly explain why they chose P8 for the viral transduction experiment. The HC differentiation rate is actually quite promising (71%). It would be interesting to see whether reprograming P14 SCs can achieve a similar efficiency. Have the authors already attempted this experiment? If yes, the results should be discussed.

3) *Atoh1*-GFP+ postnatal day 1 (P1) cochlear hair cells were sorted and used to identify 16 transcription factors. How about later postnatal stages of cochlear hair cell maturation? Would that transcriptional information be helpful to make more mature/functional hair cells?

On the other hand, identifying transcription factors in *atoh1*+ cells assumes that *atoh1* is the very first hair cell specifying transcription factor. Maybe there are transcription factors upstream of *atoh1* that would improve hair cell induction rates?

4) The authors discuss the fact that the gene expression profiles of iHCs cannot be compared to nascent vestibular or cochlear hair cell subtypes because vestibular and cochlear hair cell subtypes are too similar, which precludes a comparison with the goal of a statistically supported classification. The Discussion: "… we are unable to statistically classify…" is misleading because it suggests that the authors have attempted such a comparison. It would in fact be very interesting if the authors have performed this comparison and the results are indeed non-conclusive or, if a comparison was not attempted, the statement should be removed and replaced with a discussion how iHCs might compare with embryonic stem cell-generated or inner ear organoid-derived hair cells.

5) 6 month old adult mouse tail-tip fibroblasts only have a 0.006% conversion rate. Could the authors speculate as to how reprogramming efficiency could be improved? The morphology of iHCs derived from adult tail-tip fibroblasts is also distinct from the other cell types obtained in this study. They were sparse individual cells. However, iHCs derived from both MEFs and P8 SCs formed clusters. Human somatic cells will be the most translational target for reprograming, thus, increasing efficiency will be critical. Some discussion of this is warranted.

6) Subsection “Induced Hair Cells resemble primary juvenile mouse hair cells transcriptionally”, last paragraph: lack of expression of Zfp503 is interesting. Would addition of this TF result in increased maturity? i.e. more HC genes and fewer MEF genes but without increase of reprograming efficiency as measured for Figure 1/Figure 1—figure supplement 1. Zfp503 could be a candidate for increasing reprograming at a different level compared to the four other TFs.

The Discussion does not pick up the finding that Zfp503 expression is not detected in iHCs. It would be informative to read more about whether adding this factor would potentially improve the maturity of reprogrammed cells; particularly because the rationale why Gata3 would be a candidate does not appear to be clear in the Discussion (subsection “Reprogramming efficiency and the level of maturity of iHCs”, third paragraph).

7) Co-culture of sorted *Atoh1*-GFP+ iHCs and dissociated wildtype P1 organ of Corti:

• Have the authors tried to co-culture iHCs with intact P1 cochlear explant culture? Have the authors tried to co-culture iHCs with dissociated P1 supporting cells (i.e. by sorting out the native hair cells)? Some discussion of these alternatives would be helpful.

• From a technical standpoint, the authors should provide a better definition of "Organ of Corti". What specific cell types are included in this culture? Is there any consistency to the types of supporting cells that end up surrounding iHCs?

8) Screening approach: The authors should provide more detail regarding the automated segmentation software used for quantification. Based on the images in Figure 7B, it is difficult to discern native HCs from iHCs. How did they go about benchmarking their analysis pipeline? Perhaps some thresholded images from various stages of the pipeline would lend confidence to the reader.

---

## [Author Response]

Essential revisions:The following comments should be addressed by further experimentation or analysis:1) It should be evaluated, based on published data, whether the 16 transcription factors initially identified are also expressed by vestibular hair cells. A comparison of iHCs with cochlear hair cells is appropriate based on the rationale presented in the study, but further comparison with vestibular hair cells might reveal an even higher level of similarity.

The 16 transcription factors initially identified are all expressed by both P1 cochlear and utricular hair cells, with the exception of Zfp503. The transcription factor Zfp503 is only expressed in the cochlear hair cells. Based on this expression, the lack of Zfp503 in the iHCs may suggest that the lack of this transcription factor is making them more vestibular in nature, however as supported by the PCA plot (Figure 2—figure supplement 1A), the iHCs are not more similar to P1 cHCs or P1 uHCs, at a bulk transcriptional level. Future single cell analysis, including the addition of Zfp503 and other transcription factors to our reprogramming cocktail, will better determine the necessary components in the differentiation of specified hair cell types from these two tissues, as well as those components that are needed for maturation of specific hair cell subtypes.

A figure showing the comparison of iHCs to P1 primary cochlear hair cells and P1 utricular hair cells by PCA analysis, as well as a figure showing the expression levels of the 12 additional reprogramming transcription factors tested alongside *Six1*, *Atoh1*, *Pou4f3* and *Gfi1* has been added to Figure 2—figure supplement 1A, D and discussed in the Results (subsection “Induced Hair Cells resemble primary juvenile mouse hair cells transcriptionally”) and the Discussion (subsection “Reprogramming efficiency and the level of maturity of iHCs”). The expression bar graphs (Figure 2—figure supplement 1A and Figure 2D) are now represented in Rlog counts instead of FPKM, as we now believe that FPKM from gene level quantification is complicated by single-end versus paired-end sequencing and differences in library preparation. We have now included the expression level of the 12 additional reprogramming factors as Rlog counts instead (Figure 2—figure supplement 1D). The data added to the supplementary figure indicates the similarity of iHCs to the two primary hair cell types. By noting the variance in the PCA, it is clear that induced hair cells are not more like one than the other at this stage of maturation.

We thank our reviewer for suggesting we include these data and explanation.

2) FM dye uptake assay: The images in Figure 6C are not entirely convincing. For instance, it would aid with the interpretation to see FM uptake in a native hair cell vs. iHCs in this co-culture setup.

We agree with the reviewer, and while our data clearly indicate a rapid and selective uptake of FM4-64 into reprogrammed hair cells, we acknowledge that the pictures are not optimal. The challenge has been the accumulation of cell debris in the monolayer culture of mouse embryonic fibroblasts from which the induced hair cells are derived. Both the live nuclear dye (NucBlue), and the FM4-64, rapidly label the debris associated with these cultures. We undertook to redo these experiments in the context of our co-cultures, with the hope that induced hair cells could be monitored alongside primary hair cells embedded in the epithelial islands that form (Figure 6 and Figure 6—figure supplement 1). Unfortunately, these experiments were interrupted by the Corona virus shutdown just as we were beginning to set up the cultures.

In lieu of new experiments, we have gone back to the original images for Figure 6C, and using ImageJ, we cropped a smaller area and filtered the images on the “minimum background intensity”. This filter measures the minimum signal intensity found in the image and applies the filter to remove the minimum signal across the entire image. This modification step was used uniformly on all channels to ensure that the filtering was not changing the image in a biased way. We were also able to include an image of the same FM treatment, performed on dissociated primary hair cells from P1 transgenic mice. The primary hair cells express the *Atoh1*::nGFP reporter and accumulate the FM dye in the same time frame (after a 30 second treatment) as the induced hair cells. The same scale of 25µm and the same minimum background intensity filter was used in the iHC image and the primary HC image. We feel that this change improves the quality of our images, and hope that the reviewers find these changes sufficient to clearly demonstrate the rapid uptake of dye in induced hair cells.

We have taken this opportunity to reorganize Figure 6. The portions of text corresponding to the FM images have been moved to the subsection “Induced hair cells demonstrate voltage-dependent ion currents”, so it comes after the electrophysiology portion of the text. This also led us to relabel the figure panels so that they match the modified Figure 6. The new Figure 6 has been provided and the figure legend changed accordingly. The Materials and methods section was also changed accordingly.

3) The signature genes for Merkel Cells omit some classic markers, such as Sox2, Isl1, Krt20. Is it possible that this dataset contains a mixture of keratinocytes and Merkel cells?

With regard to the GSEA analysis, this was constructed of genes that were unique to each individual cell type analyzed (MEF, HC, MC, GUT, CGP), thus excluding those genes that are expressed in any two cell types. The genes *Sox2* and Isl1 are expressed in Merkel cells, but also expressed in primary hair cells, thus they were eliminated from the cell specific gene sets. Similarly, the gene Krt20, while a characteristic Merkel cell marker, is also expressed in the gut secretory cells. The gene-lists generated for GSEA analysis were intended to represent genes specific to only one of the cell types, for the purposes of comparing the multiple cell types. This allowed an overall comparison between induced hair cells, with MEF, HC, MC, CGP, and GUT as a group, and allowed better statistical analysis.

We have changed the text to include the following sentence: “The GSEA program identified gene-lists exclusive to each cell type; these gene-lists included only genes which were not expressed in any two cell types.”

The following comments should be discussed:1) A powerful experiment that would suggest translatability would be to add the cocktail to rodent utricle cultures treated with an aminoglycoside. Have the authors already attempted this experiment? If yes, the results should be discussed.

The above experiment has not been attempted yet, but it is on the list. We are working to develop a minimum number of polycistronic AAV constructs for our transcription factor combination of SAPG. AAV delivery has only recently been successful in the context of the inner ear, and will allow us to pursue reprogramming in organ explants of the organ of Corti or utricle, as well as in vivo. Our future experiments will consist of ablating some or all of the primary hair cells, using either aminoglycosides or DTR mice. Following ablation, we will introduce the reprogramming transcription factors to see if we can convert the primary supporting cells to a hair cell-like fate in the context of the intact organ of Corti or utricle, both in vitro and in vivo. We have modified the Discussion to include suggestions for this line of research.

2) Postnatal day 8 (P8) cochlear supporting cells: The author explain that supporting cells became an interesting target to drive HC differentiation, but they do not clearly explain why they chose P8 for the viral transduction experiment. The HC differentiation rate is actually quite promising (71%). It would be interesting to see whether reprograming P14 SCs can achieve a similar efficiency. Have the authors already attempted this experiment? If yes, the results should be discussed.

We have not attempted to perform this experiment using P14 SCs, due to the technical challenges associated with dissecting, dissociating, culturing and FACS-sorting the inner ear at later post-natal stages. There is a latent ability of perinatal organ of Corti supporting cells to transdifferentiate in response to either Notch inhibition or *Atoh1* over expression (Kelly et al., 2012; Maass et al., 2015). This ability is present at P1 (White et al., 2006), but has disappeared by P6. P8 was chosen as the latest time point when the temporal bone is still soft enough to allow reliable preparation of FACS-purified supporting cells. By using P8 supporting cells as our starting material for viral transduction with the reprogramming transcription factors, we have minimized the possibility for spontaneous transdifferentiation in the experiment.

We have modified the subsection “*Six1*, *Atoh1*, *Pou4f3*, and *Gfi1* are capable of reprogramming postnatal and adult somatic cells” to explain the use of the developmental stage of P8 supporting cells.

3) Atoh1-GFP+ postnatal day1 (P1) cochlear hair cells were sorted and used to identify 16 transcription factors. How about later postnatal stages of cochlear hair cell maturation? Would that transcriptional information be helpful to make more mature/functional hair cells?

At the time these experiments were initiated, we did not have reliable transcriptomes of older hair cells, which are notoriously hard to profile due to the difficulty of acquiring sufficient numbers of FACS-pure, viable cells. Also, we felt that P1 represented a good target population, as being definitive hair cells that will not revert to a progenitor phenotype if *Atoh1* is knocked out (Maass et al., 2015). Thus, this *Atoh1*-independence suggests that if we could get them to this stage, subsequent maturation would be possible based on an initial gene hierarchy, dependent on *Atoh1*. Of course, this was guess work, but it represents our reasoning at a time when nothing was known about the possibility of reprogramming unrelated somatic cells to a hair cell fate.

In the event, and as we have discussed, the induced hair cells that we have produced are not fully mature, and indeed, engraftment with primary hair cells and supporting cells in our co-cultures (Figure 6 and Figure 6—figure supplement 1) only modestly improves their molecular maturity. We agree that it is likely that additional transcription factors will be needed to bring about maturity, and mining our, now available, older transcriptomes has been ongoing. In the next iteration of these experiments, we will test several additional transcription factors that we hope will induce a more mature cell type during the reprogramming process. We have added this suggestion to the Discussion.

On the other hand, identifying transcription factors in atoh1+ cells assumes that atoh1 is the very first hair cell specifying transcription factor. Maybe there are transcription factors upstream of atoh1 that would improve hair cell induction rates?

We agree with this comment, and indeed, our transcription factor cocktail includes *Six1*, which is expressed upstream of *Atoh1* and may contribute to preparing the epigenetic landscape of the sensory primordia. This hypothesis will be tested in the context of single-cell analysis of reprogramming, that we planned before the Corona virus shut us down.

The other reason that we agree with this idea, is that we have discovered that although *Atoh1* is technically epistatic to *Pou4f3*, *Atoh1* requires *Pou4f3* activity to access much of its targetome (in preparation). Thus, our unbiased choice of reprogramming factors identified two factors, *Atoh1* and *Pou4f3*, that synergize to drive expression of the hair cell phenotype. We imagine that there are additional synergies that are likely to be important during hair cell maturation, and that using newly engineered reporters for more mature markers, will enable a similarly unbiased selection of TFs that act at lower levels in the hair cell gene regulatory network.

We thank the reviewer for this query and have added a short sentence in the discussion to explain that *Six1* expression precedes *Atoh1* activation in the developing organ of Corti (subsection “Reprogramming efficiency and the level of maturity of iHCs”).

4) The authors discuss the fact that the gene expression profiles of iHCs cannot be compared to nascent vestibular or cochlear hair cell subtypes because vestibular and cochlear hair cell subtypes are too similar, which precludes a comparison with the goal of a statistically supported classification. The Discussion: "… we are unable to statistically classify…" is misleading because it suggests that the authors have attempted such a comparison. It would in fact be very interesting if the authors have performed this comparison and the results are indeed non-conclusive or, if a comparison was not attempted, the statement should be removed and replaced with a discussion how iHCs might compare with embryonic stem cell-generated or inner ear organoid-derived hair cells.

We have in fact performed the comparison of iHCs with P1 cochlear hair cells and P1 utricular hair cells, and have now included this analysis as a PCA plot in Figure 2—figure supplement 1A) (as described in our response to “Essential revision 1” above). This change has been discussed in the Results (subsection “Induced Hair Cells resemble primary juvenile mouse hair cells transcriptionally”) and the Discussion (subsection “Reprogramming efficiency and the level of maturity of iHCs”).0

The GSEA comparison was done only in the context of deriving a signature gene set. We did not mean to imply that utricular hair cells were indistinguishable from cochlear hair cells. They have large transcriptional differences. However, for the goal of selecting genes that demonstrate exclusive expression across all *Atoh1* positive cell-types in this study, utricular and cochlear hair cells differed primarily by intensity of expression, not exclusiveness.

5) 6 month old adult mouse tail-tip fibroblasts only have a 0.006% conversion rate. Could the authors speculate as to how reprogramming efficiency could be improved? The morphology of iHCs derived from adult tail-tip fibroblasts is also distinct from the other cell types obtained in this study. They were sparse individual cells. However, iHCs derived from both MEFs and P8 SCs formed clusters. Human somatic cells will be the most translational target for reprograming, thus, increasing efficiency will be critical. Some discussion of this is warranted.

Tail-tip fibroblasts (TTFs) are typically used in reprogramming studies as a proof-of-principle that mature somatic cell types from alternate embryonic lineages can be reprogrammed. Their reprogramming has been shown to be much less efficient, regardless of the target cell type and commonly require additional medium supplementation (Liu et al., 2011; Lalit et al., 2016). In addition, they are difficult to culture, expand, and infect. The extremely low conversion rate reported in the manuscript is typical of this cell type in many other reprogramming protocols regardless of target cell type. We believe that the low conversion rate also contributed to the difference in morphology, since there were simply not sufficient reprogrammed cells to induce the kind of clustering seen with the MEFS. In addition, it is likely that the epigenetic landscape of the TTF is significantly different than the less mature MEFS, and certainly than the more highly-related P8 supporting cells. We hope that this will, in the future, contribute to the success of in vivo reprogramming, which from a translational perspective, is one of our long-term goals. Indeed, we speculate that unlike TTF, surviving supporting cells in long-deafened mice and humans, are likely to have an epigenetic memory that is more similar to hair cells, which share an immediate developmental lineage, as compared to a distantly related cell type like TTFs. We thank the reviewer, and have modified the text to include this issue in the subsection “*Six1*, *Atoh1*, *Pou4f3*, and *Gfi1* are capable of reprogramming postnatal and adult somatic cells”.

6) Subsection “Induced Hair Cells resemble primary juvenile mouse hair cells transcriptionally”, last paragraph: lack of expression of Zfp503 is interesting. Would addition of this TF result in increased maturity? i.e. more HC genes and fewer MEF genes but without increase of reprograming efficiency as measured for Figure 1/Figure 1—figure supplement 1. Zfp503 could be a candidate for increasing reprograming at a different level compared to the four other TFs.The Discussion does not pick up the finding that Zfp503 expression is not detected in iHCs. It would be informative to read more about whether adding this factor would potentially improve the maturity of reprogrammed cells; particularly because the rationale why Gata3 would be a candidate does not appear to be clear in the Discussion (subsection “Reprogramming efficiency and the level of maturity of iHCs”, third paragraph).

Zfp503 is a transcription factor highly expressed in P1 cochlear hair cells, compared to P1 utricular hair cells. It is possible that the lack of Zfp503 in the iHCs suggests a lack of cochlear specification, however this is unclear given our bulk RNA sequencing results, as discussed in the response to Question 1 of the “Essential Revisions”. The addition of Zfp503 to the SAPG transcription factor cocktail cut the reporter activation in half (see Figure 1—figure supplement 1G), and so for the purposes of this first-generation reprogramming protocol, was not used in the final cocktail. However, we agree with our reviewer that it’s inclusion in subsequent experiments could be extremely interesting with regard to HC maturation, as well as inner vs. outer differentiation. The next iteration of these experiments will be analyzed using single cell RNAseq and ATACseq to analyze additional TFs that are suggested by our bioinformatic analysis. We have modified the text in the Discussion to include a brief discussion of this gene.

7) Co-culture of sorted Atoh1-GFP+ iHCs and dissociated wildtype P1 organ of Corti:• Have the authors tried to co-culture iHCs with intact P1 cochlear explant culture? Have the authors tried to co-culture iHCs with dissociated P1 supporting cells (i.e. by sorting out the native hair cells)? Some discussion of these alternatives would be helpful.• From a technical standpoint, the authors should provide a better definition of "Organ of Corti". What specific cell types are included in this culture? Is there any consistency to the types of supporting cells that end up surrounding iHCs?

Organ of Corti used in co-cultures is an epithelial preparation consisting of the sensory epithelium with a small portion of surrounding mesenchyme (described in Materials and methods). The presence of dissociated periotic mesenchyme in the cocultures is essential for the self-organization into epithelial islands (Doetzlhofer et al., 2004; White et al., 2006). We tested the purification and addition of mesenchyme and sensory epithelium separately, but found that doing a preparation with both provided more consistent co-cultures, for reasons that are not clear. We have modified a sentence to clearly state the cell types added during the co-culture protocol in the subsection “Morphological characterization of induced hair cells”.

8) Screening approach: The authors should provide more detail regarding the automated segmentation software used for quantification. Based on the images in Figure 7B, it is difficult to discern native HCs from iHCs. How did they go about benchmarking their analysis pipeline? Perhaps some thresholded images from various stages of the pipeline would lend confidence to the reader.

The experiments in Figure 7 describing our ototoxicity experiments were conducted on reprogrammed iHCs from MEFs, and not co-cultures, so no primary HCs were present. The survival assays on iHCs and primary hair cells were done independently from each other, since both systems employ the same *Atoh1*::nGFP reporter. The inclusion of primary hair cell cultures was done for comparison only. The rationale behind the experiment is to produce an efficient assay that could be used for high-throughput drug screening. Addition of primary cells from the organ of Corti would induce epithelial island formation, which is harder to image, and thus confound this goal.

Automated cell counting used thresholds for size, intensity and roundness of the *Atoh1*::nGFP signal (as described in the Materials and methods). Since the GFP was localized to the nucleus in the induced hair cells, as well as the primary hair cells, the same quantification parameters were used. The imaging is done at 10x and the nuclei count recipe parameters include: background removal factor (removes variation in background), contrast thresholding (allows user to adjust detection sensitivity), smoothing factor (smooths nuclei boundaries), minimum and maximum object size (objects not within the specified size range will be removed), and a separation filter (separates neighboring objects). All of these parameters were optimized using reference 10x images of the *Atoh1*::nGFP reporter. A p-value is reported in the subsection “Induced hair cells recapitulate sensitivity to gentamicin, a known ototoxin”, which represents a comparison of 20 wells counted manually versus automatically.

Author response image 1 is a representative image of one well of a 96 well plates. The well is imaged as 16 separate tiles and then stitched together. If the reviewers wish to see this added to the supplementary data we can include it.
